# Improving Group Connectivity for Generalization of Federated Deep Learning

## Abstract

Federated learning (FL) involves multiple heterogeneous clients collaboratively training a global model via iterative local updates and model fusion. The generalization of FL's global model has a large gap compared with centralized training, which is its bottleneck for broader applications. In this paper, we study and improve FL's generalization through a fundamental "connectivity" perspective, which means how the local models are connected in the parameter region and fused into a generalized global model. The term "connectivity" is derived from linear mode connectivity (LMC), studying the interpolated loss landscape of two different solutions (e.g., modes) of neural networks. Bridging the gap between LMC and FL, in this paper, we leverage fixed anchor models to empirically and theoretically study the transitivity property of connectivity from two models (LMC) to a group of models (model fusion in FL). Based on the findings, we propose FedGuCci(+), improving group connectivity for better generalization. It is shown that our methods can boost the generalization of FL under client heterogeneity across various tasks (4 CV datasets and 6 NLP datasets) and model architectures (e.g., ViTs and PLMs).

## 1 Introduction

Federated learning (FL) is a privacy-preserving and communication-efficient distributed training paradigm that enables multiple data owners to collaboratively train a global model without sharing their data (McMahan et al., 2017). However, clients always have heterogeneous data (Li et al., 2020a; Lin et al., 2020), and in each round, they conduct local training of multiple epochs based on the data, causing model drifts of local models (Karimireddy et al., 2020; Wang et al., 2020), further resulting in generalization degradation of the fused global model (Li et al., 2023a; Acar et al., 2020). Previous works improve the generalization by seeking flatter minima (Caldarola et al., 2022; Qu et al., 2022) or using local proximal regularization (Li et al., 2020a) to remedy the model drifts. While in this paper, we take a more fundamental perspective on how the local models are **connected** with each other under model drifts (**group connectivity**) and how they are fused into a **generalized** global model based on such connectivity.

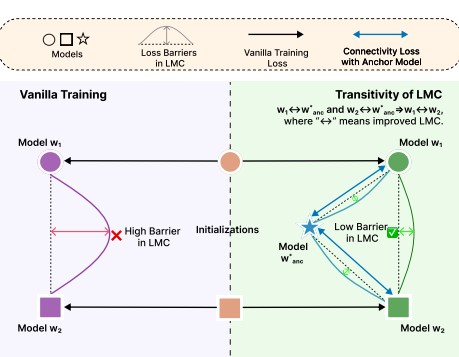

Figure 1: **Illustration on transitivity of linear mode connectivity.** Left: vanilla training, where models have high barriers in LMC. Right: transitivity of LMC. Models $\mathbf{w}_1$ and $\mathbf{w}_2$ are independently trained, and they are all learned to have good LMC with anchor model $\mathbf{w}_{\text{anc}}^*$. At the end of the training, models $\mathbf{w}_1$ and $\mathbf{w}_2$ have improved LMC, showing the transitivity of LMC.

The notion of group connectivity is inspired by linear mode connectivity (LMC), which studies the interpolated loss landscape of two SGD solutions (e.g., modes) (Draxler et al., 2018; Zhang et al., 2021; Entezari et al., 2022). It is found that two trained models with different random seeds of batch orders (depicted by *SGD noise*), even if have the *same initialization*, may cause a barrier along their *linear interpolation path* (i.e., the LMC path), indicating the two SGD solutions are not in the same loss landscape basin (Draxler et al., 2018; Garipov et al., 2018; Ainsworth et al., 2022). This observation is quite analogous to model drift in FL, where multiple local models are *initialized the same*, but due to *SGD noise and bias* (Li et al., 2020a; Karimireddy et al., 2020) caused by heterogeneous data

and asynchronous training, local models drift from each other and have inferior generalization after *linear model fusion*. This analogy inspires us to think about whether we can leverage the insights and techniques from LMC to improve the generalization of FL through the lens of connectivity. Previous works propose to learn neural network subspaces for increasing LMC between two models when simultaneously training them (Wortsman et al., 2021; Garipov et al., 2018). They use the midpoints of the improved LMC for ensembling. In this paper, we aim to leverage the idea of increasing LMC to improve the connectivity among the local models in FL. However, there is a crucial gap between LMC and FL. In Wortsman et al. (2021), they can retain and train two models simultaneously, while in each round of FL, every local model is independently trained for several epochs. In addition, LMC only considers two models, while FL requires the connectivity of multiple models.

Therefore, we utilize a fixed anchor model to study the transitivity property of LMC and hypothesize that: if LMC between model $\mathbf{w}_1$ and anchor model $\mathbf{w}_{\text{anc}}^*$, as well as between model $\mathbf{w}_2$ and anchor model $\mathbf{w}_{\text{anc}}^*$, is independently enhanced, then the LMC between models $\mathbf{w}_1$ and $\mathbf{w}_2$ will also improve (an illustration of the transitivity is in Figure 1). Through theoretical and empirical analyses, we verify the transitivity of LMC and then extend it to the group connectivity of multiple models.

Based on the above findings, we propose **Fed**erated Learning with Improved **Gr**ou**p** **C**onne**c**tiv**i**ty (**FedGuCci**), which leverage the global models as the anchor models for improving group connectivity of local models. Further, due to data heterogeneity in FL, clients' local loss landscapes are different and shifted. Thus, we propose a strengthened version, FedGuCci+, by incorporating some heterogeneity-resistant modules for aligning local loss landscapes. Our contributions are listed below.

- We study FL from the connectivity perspective, which is novel and fundamental to understanding the generalization of FL's global model.
- We theoretically and empirically verify the transitivity of LMC and the group connectivity of multiple models.
- We propose FedGuCci and FedGuCci+. Extensive experiments show that our methods can improve the generalization of FL across various settings.

The rest of the paper is organized as follows. In section 2, we provide the preliminaries of FL and LMC and the most related works. In section 3, we give the hypothesis about the transitivity of connectivity and the theoretical and empirical analyses. Based on the findings, in section 4, we propose FedGuCci(+) in FL, and then the experimental results are in section 5. Lastly, we conclude the paper in section 6.

## 2 PRELIMINARIES AND RELATED WORKS

In this section, we present the preliminaries of FL and LMC and the most relevant works to this paper.

### 2.1 PRELIMINARY OF FEDERATED LEARNING

FL includes a server and $M$ clients to collaboratively learn a global model without data sharing (McMahan et al., 2017). Denote the set of clients by $\mathcal{S}$, the local dataset of client $i$ by $\mathcal{D}_i = \{(x_j, y_j)\}_{j=1}^{|\mathcal{D}_i|}$, the sum of clients' data by $\mathcal{D} = \bigcup_{i \in \mathcal{S}} \mathcal{D}_i$. The IID data distributions of clients refer to each client's distribution $\mathcal{D}_i$ is IID sampled from $\mathcal{D}$. However, in practical FL scenarios, *heterogeneity* exists among clients whose data are *non-IID* with each other, causing model drifts. During FL training, clients iteratively conduct local updates and communicate with the server for model fusion. In the local updates, *the number of local epochs is $E$*; when $E$ is larger, the communication is more efficient, but the updates are more asynchronous, also the model drifts are more severe. The total number of communication rounds is $T$.

Denote the global model and the client $i$'s local model in communication round $t \in [T]$ by $\mathbf{w}_g^t$ and $\mathbf{w}_i^t$. In each round, clients' local models are initialized as the global model that $\mathbf{w}_i^t \leftarrow \mathbf{w}_g^t$, and clients conduct local training in parallel. In each local training epoch, clients conduct SGD update with a local learning rate $\eta_l$, and each SGD iteration shows as

$$\mathbf{w}_i^t \leftarrow \mathbf{w}_i^t - \eta_l \nabla \ell(B_b, \mathbf{w}_i^t), \text{ for } b = 1, 2, \cdots, B, \tag{1}$$

where $\ell$ is the batch-level loss function and $B_b$ is the mini-batch sampled from $\mathcal{D}_i$ at the $b$-th iteration. After local updates, the server samples a set $\mathcal{S}^t$ of $K$ clients and conducts *linear model fusion* to

generate a new global model. The participation ratio is $\rho = \frac{K}{M}$. The model fusion process is as

$$\mathbf{w}_g^{t+1} = \sum_{i \in \mathcal{S}^t} \mu_i \mathbf{w}_i^t, \text{ s.t. } \mu_i \geq 0, \tag{2}$$

where $\boldsymbol{\mu} = [\mu_i]_{i \in \mathcal{S}^t}$ is the fusion weights. For vanilla FedAvg, it adopts normalized weights proportional to the data sizes, $\mu_i = \frac{|\mathcal{D}_i|}{|\mathcal{D}^t|}, \mathcal{D} = \bigcup_{i \in \mathcal{S}^t} \mathcal{D}_i$. A recent study shows that the sum of fusion weights can be smaller than 1 to improve generalization by global weight decay regularization (Li et al., 2023a).

## 2.2 PRELIMINARY OF LINEAR MODE CONNECTIVITY

**Linear mode connectivity (LMC).** LMC refers to the loss landscape where two models $\mathbf{w}_1$ and $\mathbf{w}_2$ are linearly interpolated by $\mathbf{w} = \alpha\mathbf{w}_1 + (1-\alpha)\mathbf{w}_2$, for $\alpha \in [0,1]$. Usually, there are three forms of LMC regarding different $\mathbf{w}_1$ and $\mathbf{w}_2$. (1) LMC between two SGD solutions with the same initialization but different random seeds (batch orders) (Ainsworth et al., 2022); (2) LMC between two SGD solutions with different initializations (Entezari et al., 2022); (3) LMC from the initialization and the trained model (Vlaar & Frankle, 2022). LMC is depicted by the barriers in the landscape, the lower the barriers, the better the LMC. We present the definitions of loss and accuracy barriers below.

**Definition 2.1** *Loss and accuracy barriers. Let $f_{\mathbf{w}}(\cdot)$ be a function represented by a neural network with parameter vector $\mathbf{w}$ that includes all parameters. $\mathcal{L}(\mathbf{w})$ is the given loss (e.g., train or test error) of $f_{\mathbf{w}}(\cdot)$ and $\mathcal{A}(\mathbf{w})$ is its accuracy function. Given two independently trained networks $\mathbf{w}_1$ and $\mathbf{w}_2$, let $\mathcal{L}(\alpha\mathbf{w}_1 + (1-\alpha)\mathbf{w}_2)$ be the averaged loss of the linearly interpolated network and $\mathcal{A}(\alpha\mathbf{w}_1 + (1-\alpha)\mathbf{w}_2)$ be its accuracy, for $\alpha \in [0,1]$. The loss barrier $B_{loss}(\mathbf{w}_1, \mathbf{w}_2)$ and accuracy barrier $B_{acc}(\mathbf{w}_1, \mathbf{w}_2)$ along the linear path between $\mathbf{w}_1$ and $\mathbf{w}_2$ are defined as:*

$$B_{loss}(\mathbf{w}_1, \mathbf{w}_2) = \sup_{\alpha} \left\{ [\mathcal{L}(\alpha\mathbf{w}_1 + (1-\alpha)\mathbf{w}_2)] - [\alpha\mathcal{L}(\mathbf{w}_1) + (1-\alpha)\mathcal{L}(\mathbf{w}_2)] \right\}. \tag{3}$$

$$B_{acc}(\mathbf{w}_1, \mathbf{w}_2) = \sup_{\alpha} \left[ 1 - \frac{\mathcal{A}(\alpha\mathbf{w}_1 + (1-\alpha)\mathbf{w}_2)}{\alpha\mathcal{A}(\mathbf{w}_1) + (1-\alpha)\mathcal{A}(\mathbf{w}_2)} \right]. \tag{4}$$

The loss barrier is not bounded, while the accuracy barrier is bounded within $[0,1]$.

**Reducing the barriers in LMC.** In Wortsman et al. (2021), the authors train two SGD solutions simultaneously while also learning a line of connected subspace between the two models. It also adds a regularization loss to make the two solutions orthogonal so that the midpoints of the LMC path can have diversity for ensembling. While in our paper, we also use similar techniques for improving LMC, but we do not require orthogonality. Also, instead of simultaneously training two models, we individually train models, improve their LMC with a fixed anchor model, and verify the LMC's transitivity.

## 2.3 MOST RELATED WORKS

**LMC and FL.** In Hahn et al. (2022), the authors propose to train two models (one for personalization and another for generalization) at clients and learn a connected subspace between the two models for better personalization. Recently, a concurrent work (Zhou et al., 2023) empirically and theoretically verifies that when clients' data are more heterogeneous, the local loss landscapes will be more shifted, causing worse LMC. However, they haven't proposed an effective algorithm in FL based on LMC insights, where our contributions lie. To the best of our knowledge, our paper may be the first paper to study and improve the generalization of FL from the connectivity perspective.

**Comparison with FedProx.** FedProx (Li et al., 2020a) adopts the current round's global model as a regularization term for tackling heterogeneity. Instead, we utilize the historical global models as the anchor models and learn to improve the connectivity between the local model with these anchor models. Thus, our methods and FedProx have fundamental differences in leveraging the global models regarding motivation and implementations. Due to space limits, we include more related works in Appendix D, e.g., generalization of FL and LMC basics.

## 3 TOWARDS THE TRANSITIVITY OF CONNECTIVITY

In this section, we verify the transitivity of LMC and group connectivity by leveraging fixed anchor models, paving the way for improving generalization in FL.

## 3.1 TRANSITIVITY OF LINEAR MODE CONNECTIVITY

We first give the hypothesis on the transitivity of LMC.

**Hypothesis 3.1** *Transitivity of linear mode connectivity (informal).* *There are three models* $\{\mathbf{w}_1, \mathbf{w}_2, \mathbf{w}_{anc}^*\}$. *If the linear mode connectivity between* $\mathbf{w}_1$ *and* $\mathbf{w}_{anc}^*$, *as well as the one between* $\mathbf{w}_2$ *and* $\mathbf{w}_{anc}^*$, *are independently improved, then, the linear mode connectivity between* $\mathbf{w}_1$ *and* $\mathbf{w}_2$ *is also improved.*

We make a theoretical analysis to prove the transitivity of LMC. We make the assumption below, following Assumption 7 in Ferbach et al. (2023) and Assumption 1 in Li et al. (2019).

**Assumption 3.2** $\forall y \in \mathbb{Y}$, *the loss function* $L(\cdot, y)$ *is convex and 1-Lipschitz for each* $y$ *and the loss* $\mathcal{L}(\cdot)$ *is* $\gamma$-*smooth, where* $\mathcal{L}(\mathbf{w}) = \mathbb{E}[L(f_{\mathbf{w}}(x), y)]$ *and the expectation* $\mathbb{E}$ *is taken over the dataset.*

**Lemma 3.3** *Set the uniform and bounded domain for network* $\mathbf{w}$ *as* $\mathcal{E}_\epsilon = \{\mathbf{w} \in \Omega | \mathcal{L}(\mathbf{w}) < \epsilon\}$. *Define a random event* $D_\epsilon(\mathbf{w}_{anc}^*)$ *as* $D_\epsilon(\mathbf{w}_{anc}^*) = \{\exists \mathbf{w} \in \mathcal{E}_\epsilon | \forall \alpha \in [0,1], \mathcal{L}(\alpha \mathbf{w}_{anc}^* + (1-\alpha)\mathbf{w}) \leq \epsilon\}$. *Consider an anchor model* $\mathbf{w}_{anc}^*$ *and an arbitrary network* $\mathbf{w}$ *and for* $\epsilon > 0$. *For* $\|\mathbf{w} - \mathbf{w}_{anc}^*\|_\infty \leq \frac{d}{2}$,

$$P(D_\epsilon(\mathbf{w}_{anc}^*)) \leq (\frac{d_\epsilon}{d})^S, \tag{5}$$

*where* $d_\epsilon = |\mathcal{E}_\epsilon|^{\frac{1}{S}}$ *represents the average diameter of region* $\mathcal{E}_\epsilon$, $S$ *represents the number of parameters of the neural network and the equality holds if and only if* $\mathcal{E}_\epsilon \subset \{\mathbf{w} | \|\mathbf{w} - \mathbf{w}_{anc}^*\|_\infty \leq d\}$ *is a star domain centered at* $\mathbf{w}_{anc}^*$. *Thus, when* $P(D_\epsilon(\mathbf{w}_{anc}^*))) > 1 - \delta$, *it holds* $d < \frac{d_\epsilon}{(1-\delta)^{\frac{1}{S}}}$.

**Remark 3.4** *This lemma links the distance between parameters to LMC, describing that the greater the probability of LMC (i.e., a small loss barrier) existing between the network* $\mathbf{w}$ *and the anchor model* $\mathbf{w}_{anc}^*$, *the smaller the distance should be between* $\mathbf{w}$ *and* $\mathbf{w}_{anc}^*$.

Then, we provide the following theorem.

**Theorem 3.5** *We define a two-layer neural network with ReLU activation, and the function is* $f_{\mathbf{v},\mathbf{U}}(\mathbf{x}) = \mathbf{v}^\top \sigma(\mathbf{U}\mathbf{x})$ *where* $\sigma(\cdot)$ *is the ReLU activation function.* $\mathbf{v} \in \mathbb{R}^h$ *and* $\mathbf{U} \in \mathbb{R}^{h \times l}$ *are parameters*[1] *and* $\mathbf{x} \in \mathbb{R}^l$ *is the input which is taken from* $\mathbb{X} = \{\mathbf{x} \in \mathbb{R}^l | \|\mathbf{x}\|_2 < b\}$ *uniformly. Denote the deterministic anchor model as* $\mathbf{w}_{anc}^* = \{\mathbf{U}_{anc}^*, \mathbf{v}_{anc}^*\}$, *with* $\|\mathbf{v}_{anc}^*\|_2 < d_{anc}$ *and consider two different networks* $\mathbf{w}_1, \mathbf{w}_2$ *parameterized with* $\{\mathbf{U}_1, \mathbf{v}_1\}$ *and* $\{\mathbf{U}_2, \mathbf{v}_2\}$ *respectively. Each element of* $\mathbf{U}_1$ *and* $\mathbf{U}_2$, $\mathbf{v}_1$ *and* $\mathbf{v}_2$ *is sampled from a uniform distribution centered at* $\mathbf{U}_{anc}^*$ *and* $\mathbf{v}_{anc}^*$ *with an interval length of* $d$. *If with probability* $1 - \delta$, $\sup_\alpha \mathcal{L}(\alpha \mathbf{w}_{anc}^* + (1-\alpha)\mathbf{w}_1) < \epsilon$ *and* $\sup_\alpha \mathcal{L}(\alpha \mathbf{w}_{anc}^* + (1-\alpha)\mathbf{w}_2) < \epsilon$, *then with probability* $1 - \delta$, *it has,*

$$B_{loss}(\mathbf{w}_1, \mathbf{w}_2) \leq \frac{\sqrt{2h}b}{2(1-\delta)^{\frac{2}{hl+h}}} d_\epsilon(d_\epsilon + d_{anc}) \log(12h/\delta), \tag{6}$$

*where* $B_{loss}(\mathbf{w}_1, \mathbf{w}_2)$ *is the loss barrier as* Equation 3.

The proofs are in Appendix B. Theorem 3.5 proves the transitivity of LMC that when $\mathbf{w}_1$ and $\mathbf{w}_2$ have lower LMC barrier with $\mathbf{w}_{anc}^*$ (the barrier proxy is $\epsilon$) then the barrier between $\mathbf{w}_1$ and $\mathbf{w}_2$ is also reduced and bounded.

***Then, we will empirically validate the transitivity.*** We first present the connectivity loss given the anchor model, which is similar to previous literature (Wortsman et al., 2021; Garipov et al., 2018). The connectivity loss is as follows,

$$\mathcal{L}_{\text{connect}}(\mathbf{w}, \mathbf{w}_{\text{anc}}^*) = \mathbb{E}_{\alpha \sim [0,1]} \mathcal{L}(\alpha\mathbf{w} + (1-\alpha)\mathbf{w}_{\text{anc}}^*) = \int_0^1 \mathcal{L}(\alpha\mathbf{w} + (1-\alpha)\mathbf{w}_{\text{anc}}^*)\, d\alpha, \tag{7}$$

---

[1] For simplicity and without loss of generality, we omit the bias terms.

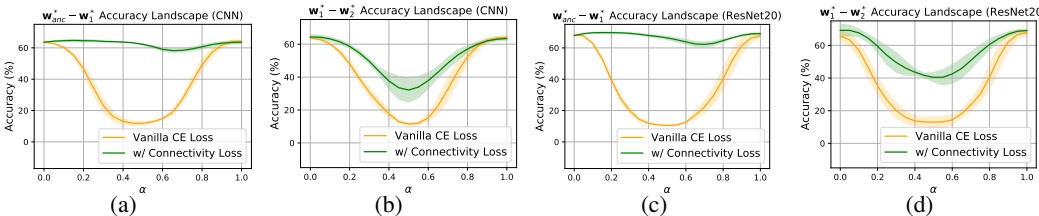

(a)  (b)  (c)  (d)

Figure 2: **Linear mode connectivity landscapes of test accuracy, showcasing the transitivity.** The accuracy barrier is shown as the maximal accuracy drop along the landscape. **(a) and (c):** LMC between one trained model and the anchor model, and the barrier is eliminated for connectivity loss. **(b) and (d):** LMC between two trained models, connectivity loss has the lower barriers, showing the transitivity of LMC. CIFAR-10 is used.

where $\mathbf{w}_{anc}^*$ is the fixed anchor model and $\mathbf{w}$ is the model for training. Then, we incorporate the connectivity loss into the vanilla cross entropy (CE) loss, formulated into the following overall learning objective,

$$\mathbf{w}^* = \arg\min_{\mathbf{w}} \mathcal{L}(\mathbf{w}) + \beta\mathcal{L}_{connect}(\mathbf{w}, \mathbf{w}_{anc}^*), \tag{8}$$

where $\mathcal{L}(\mathbf{w})$ is the vanilla CE loss and $\beta$ is the hyperparameter controlling the strength of the connectivity loss.

We let $\mathbf{w}_{anc}^*$ be the fixed trained anchor model and independently train two models $\mathbf{w}_1^*$ and $\mathbf{w}_2^*$ according to Equation 8. According to Theorem 3.5, the $\mathbf{w}_1^*$ and $\mathbf{w}_2^*$'s LMC barriers will be reduced if the transitivity holds. Note that $\mathbf{w}_1$ and $\mathbf{w}_2$ can have the same or different initializations, and the transitivity still holds; in the experiments, we make stricter verifications by setting different initializations.

**Empirical results.** We conduct experiments in Table 1 and Figure 2. The anchor model is a mode independently trained with vanilla CE loss using a different random seed. In Table 1, training with the connectivity loss can largely reduce the barriers of LMC by utilizing the anchor model, even if two models have different initializations and never communicate with each other. More intuitive landscape visualizations are in Figure 2. It can be seen that the connectivity loss can eliminate the barrier between the anchor model and the trained model, and due to the transitivity of LMC, the barrier between the two independent models is also reduced. The experiments verify the transitivity of LMC between two models, and we will show that this transitivity can be extended to the connectivity of multiple models.

Table 1: **Test accuracies and barriers of two trained models w/ and w/o connectivity loss.** "Ind. Acc." refers to $0.5 * \mathcal{A}(\mathbf{w}_1) + 0.5 * \mathcal{A}(\mathbf{w}_2)$, and "Fused Acc." refers to $\mathcal{A}(0.5 * \mathbf{w}_1 + 0.5 * \mathbf{w}_2)$. It validates the transitivity of LMC, stating that by leveraging the anchor model, the barriers of LMC are largely reduced. CIFAR-10.

| Models | Metrics | Vanilla CE Loss | w/ Connectivity Loss |
|---|---|---|---|
| CNN | Ind. Acc. | $64.0 \pm 0.5$ | $63.9 \pm 1.4$ |
| | Fused Acc. | $11.5 \pm 0.9$ | $32.1 \pm 9.0$ |
| | Acc. Barrier | 0.821 | 0.495 (39.7% ↓) |
| ResNet 20 | Ind. Acc. | $66.7 \pm 0.9$ | $69.1 \pm 2.4$ |
| | Fused Acc. | $13.0 \pm 3.8$ | $40.5 \pm 3.5$ |
| | Acc. Barrier | 0.805 | 0.415 (44.1% ↓) |
| Pretrained ResNet18 | Ind. Acc. | $55.8 \pm 6.6$ | $64.5 \pm 0.3$ |
| | Fused Acc. | $10.0 \pm 0.0$ | $62.1 \pm 0.4$ |
| | Acc. Barrier | 0.819 | 0.038 (95.4% ↓) |

**Notes:** Our Theorem 3.5 requires no assumptions on the anchor models. Though our empirical verification in Table 1 and Figure 2 uses trained minima as anchor models, it is validated in Table 7 that the transitivity of connectivity also holds when the anchor models are less performed, e.g., random initialization.

## 3.2 TRANSITIVITY OF GROUP CONNECTIVITY

We study the group connectivity among multiple models and propose the barrier of group connectivity akin to Definition 2.1 of LMC. For brevity, we only present the definition of accuracy barriers.

**Definition 3.6** *Group connectivity. The group connectivity of model set $\{\mathbf{w}_i\}_{i=1}^K$ is depicted by the loss and accuracy barrier defined as:*

$$B_{loss}(\{\mathbf{w}_i\}_{i=1}^K) = \mathcal{L}(\frac{1}{K}\sum_{i=1}^K \mathbf{w}_i) - \frac{1}{K}\sum_{i=1}^K \mathcal{L}(\mathbf{w}_i), \ B_{acc}(\{\mathbf{w}_i\}_{i=1}^K) = \left[1 - \frac{\mathcal{A}(\frac{1}{K}\sum_{i=1}^K \mathbf{w}_i)}{\frac{1}{K}\sum_{i=1}^K \mathcal{A}(\mathbf{w}_i)}\right], \tag{9}$$

*where $\mathcal{L}$ is the loss and $\mathcal{A}$ is the accuracy function. A lower barrier refers to better group connectivity.*

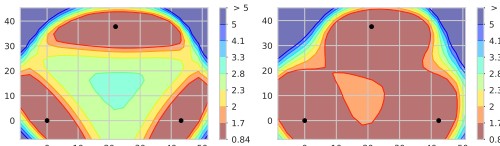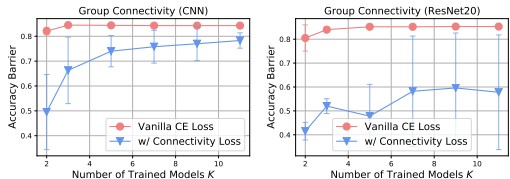

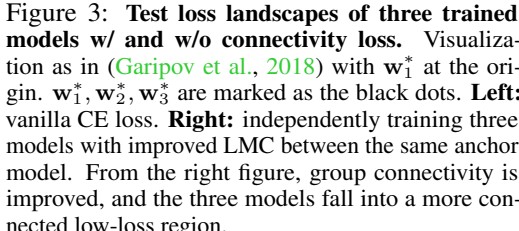

Figure 3: **Test loss landscapes of three trained models w/ and w/o connectivity loss.** Visualization as in (Garipov et al., 2018) with $\mathbf{w}_1^*$ at the origin. $\mathbf{w}_1^*, \mathbf{w}_2^*, \mathbf{w}_3^*$ are marked as the black dots. **Left:** vanilla CE loss. **Right:** independently training three models with improved LMC between the same anchor model. From the right figure, group connectivity is improved, and the three models fall into a more connected low-loss region.

Figure 4: **Accuracy barriers (the lower, the better) of group connectivity by varying numbers of trained models $K$.** There is only one anchor model for all settings. It can be seen that generally, larger $K$ will cause larger barriers, but connectivity loss can still reduce them, reflecting that the transitivity of LMC can improve group connectivity. CIFAR-10 is used.

We prove the transitivity of group connectivity that individually training several models and improving the LMC between one common anchor model will result in better group connectivity among the trained ones. In addition, we consider the data heterogeneity of practical FL in group connectivity by giving the following definition.

**Definition 3.7** *Data heterogeneity. Similar to (Li et al., 2019), we use the minimum to measure the degree of heterogeneity among the group of individual workers (e.g., clients in FL and modes in LMC). Let $\mathbf{w}^*$ be a global minimum of all workers and $\mathbf{w}_i^*$ is the minimum value of worker $i$ closest to $\mathbf{w}^*$. We use the term $\Gamma = \max_i \|\mathbf{w}_i^* - \mathbf{w}^*\|_2$, $i \in [K]$ for quantifying the degree of data heterogeneity.*

**Theorem 3.8** *We define a two-layer neural network with ReLU activation, and the function is $f_{\mathbf{v},\mathbf{U}}(\mathbf{x}) = \mathbf{v}^\top \sigma(\mathbf{U}\mathbf{x})$ where $\sigma(\cdot)$ is the ReLU activation function. $\mathbf{v} \in \mathbb{R}^h$ and $\mathbf{U} \in \mathbb{R}^{h \times l}$ are parameters and $\mathbf{x} \in \mathbb{R}^l$ is the input which is taken from $\mathbb{X} = \{\mathbf{x} \in \mathbb{R}^l | \|\mathbf{x}\|_2 < b\}$ uniformly. Denote the deterministic anchor model as $\mathbf{w}_{anc}^* = \{\mathbf{U}_{anc}^*, \mathbf{v}_{anc}^*\}$, with $\|\mathbf{v}_{anc}^*\|_2 < d_{anc}$ and consider $K$ different networks $\mathbf{w}_i$ parameterized with $\{\mathbf{U}_i, \mathbf{v}_i\}$ located on $K$ clients respectively. Each element of $\mathbf{U}_i$ and $\mathbf{v}_i$ is sampled from a uniform distribution centered at $\mathbf{U}_{anc}^*$ and $\mathbf{v}_{anc}^*$ with an interval length of d. If with probability $1 - \delta$, $\sup_\alpha \mathcal{L}_i(\alpha \mathbf{w}_{anc}^* + (1 - \alpha)\mathbf{w}_i) < \epsilon$, then with probability $1 - \delta$, it has,*

$$B_{loss}(\{\mathbf{w}_i\}_{i=1}^K) \leq \frac{\sqrt{2h}b}{2(1-\delta)^{\frac{2}{hl+h}}} d_{\epsilon+\gamma\Gamma^2}(d_{\epsilon+\gamma\Gamma^2} + d_{anc}) \log(4hK^2/\delta). \tag{10}$$

**Landscape visualization.** We empirically study whether the transitivity of LMC can be generalized to group connectivity of multiple models. We let $\mathbf{w}_{anc}^*$ be the anchor model and independently train three models $\mathbf{w}_1^*, \mathbf{w}_2^*, \mathbf{w}_3^*$ according to Equation 8. Also, training the three models without connectivity loss is conducted for comparison. Then, we visualize the loss landscapes of $\mathbf{w}_1^*, \mathbf{w}_2^*, \mathbf{w}_3^*$ in Figure 3. For vanilla CE loss, the trained models are scattered in different loss basins with high barriers between them. However, with the connectivity loss, the LMC between each model and the anchor model is improved, and as a result of transitivity, the three models fall into a more connected low-loss region, and the barriers are largely eliminated.

**Group connectivity when vary $K$.** We study the transitivity of group connectivity by scaling up the number of trained models $K$, which is critical for federated learning with numerous clients. The results are in Figure 4; note that the number of anchor models is still one. We observe that by increasing $K$ for the connectivity loss, the barrier in group connectivity will go up but still lower than the vanilla training. Also, the increase of barriers may converge to a point lower than vanilla training. It indicates that the transitivity of group connectivity may be weakened for larger $K$ but still effective, and when $K$ is relatively large (e.g., >8), increasing $K$ will cause little loss of group connectivity. Furthermore, we will show in Table 4 that our FedGuCci, which incorporates the connectivity loss, can improve the generalization under different large numbers of clients.

Table 2: **Results in terms of generalization accuracy (%) of global models on four datasets under different data heterogeneity.** The best two methods in each setting are highlighted in **bold** fonts. $M = 50, E = 3$.

| Dataset | Fashion-MNIST | | CIFAR-10 | | CIFAR-100 | | Tiny-ImageNet | |
|---|---|---|---|---|---|---|---|---|
| Non-IID hyper. | 100 | 0.5 | 100 | 0.5 | 100 | 0.5 | 100 | 0.5 |
| Local | 76.22±0.16 | 62.24±0.35 | 36.69±0.10 | 29.73±0.36 | 7.36±0.14 | 6.97±0.08 | 6.47±0.12 | 6.09±0.02 |
| FedAvg | 87.94±0.34 | 86.99±0.04 | 63.55±0.16 | 63.99±0.32 | 27.21±0.96 | 25.60±0.62 | 27.43±1.39 | 25.11±1.82 |
| FedProx | 10.00±0.00 | 10.00±0.00 | 61.81±0.47 | 61.45±0.43 | 27.78±0.41 | 28.58±0.28 | 24.58±0.28 | 25.02±0.19 |
| FedDyn | 88.26±0.17 | 88.18±0.36 | 64.99±0.64 | 65.73±0.31 | 29.90±7.13 | 28.49±0.55 | 30.89±0.03 | 24.63±2.68 |
| SCAFFOLD | 87.95±0.31 | 86.47±0.14 | 63.20±0.32 | 63.96±0.41 | 1.07±0.09 | 1.25±0.07 | 0.529±0.05 | 0.517±0.02 |
| MOON | 86.95±0.09 | 86.02±0.29 | 64.24±0.65 | 63.41±0.31 | 28.97±1.69 | 27.36±0.71 | 27.88±1.08 | 25.34±0.66 |
| FedRoD | 87.97±0.40 | 87.56±0.60 | 62.64±0.20 | 62.56±0.46 | 26.94±0.78 | 25.90±1.20 | 27.67±1.64 | 25.55±1.56 |
| FedLC | 87.90±0.36 | 86.79±0.29 | 63.49±0.17 | 63.97±0.35 | 27.23±0.69 | 25.36±0.65 | 27.63±1.62 | 25.47±1.84 |
| FedSAM | 88.41±0.49 | 87.62±0.30 | 65.10±0.41 | 65.02±0.15 | 28.11±0.61 | 26.75±0.74 | 31.23±0.16 | 30.44±0.97 |
| **FedGuCci** | **88.85±0.11** | **88.30±0.39** | **65.11±0.11** | **65.80±0.22** | **30.55±0.67** | **29.33±0.41** | **36.46±0.40** | **33.61±0.60** |
| **FedGuCci+** | **89.38±0.14** | **88.61±0.40** | **68.11±0.27** | **66.44±0.69** | **36.20±1.06** | **35.34±0.68** | **37.42±0.52** | **34.80±0.35** |

# 4 METHODS

## 4.1 FEDGUCCI: FL WITH IMPROVED GROUP CONNECTIVITY

In section 3, we have verified the transitivity of group connectivity by using an anchor model. In this section, we will present FedGuCci, incorporating this property in FL to improve generalization.

**Global models as anchor models.** We refer to subsection 2.1 for the settings and notations. In our FedGuCci, we use the global models as the anchor models for connectivity loss with local clients. Instead of solely using the current round global model as the anchor, we find using several previous rounds' global models can form the clients into a more connected region, so we use $N$ previous global models as the anchors. Specifically, in round $t \in [T]$, the set of anchor models $\mathbf{W}_{\text{anc}^*}^t$ is:

$$\mathbf{W}_{\text{anc}^*}^t = \begin{cases} \{\mathbf{w}_g^j\}_{j=t-N+1}^t & \text{if } t \geq N, \\ \{\mathbf{w}_g^j\}_{j=1}^t & \text{if } t < N, \end{cases} \tag{11}$$

where $\mathbf{w}_g^j$ refers to the global model at round $j$.

**FedGuCci local updates.** FedGuCci is a client-side algorithm that utilizes the global models as the anchor and improves the group connectivity of clients, without additional communication overhead. FedGuCci has the following update rules. In each round $t$, client $i \in [M]$ conducts local training according to the following objective:

$$\mathbf{w}_i^{t*} = \arg\min_{\mathbf{w}_i^t} \mathcal{L}_i(\mathbf{w}_i^t) + \beta \frac{1}{|\mathbf{W}_{\text{anc}^*}^t|} \sum_{j=1}^{|\mathbf{W}_{\text{anc}^*}^t|} \mathcal{L}_{\text{connect}_i}(\mathbf{w}_i^t, \mathbf{W}_{\text{anc}^*,j}^t), \tag{12}$$

where $\mathbf{W}_{\text{anc}^*,j}^t$ refers to the $j$-th model in the anchor model set, $\beta$ is the hyperparameter for connectivity loss, $\mathcal{L}_i$ is the client's local CE loss, and $\mathcal{L}_{\text{connect}_i}$ is the connectivity loss regarding Equation 7. Clients conduct SGD as Equation 1 to update the local models.

By learning to connect with the global anchor models, FedGuCci will improve the group connectivity and achieve better generalization as we will elaborate in section 5. The pseudo-code is in 1.

**Notes:** We note that our method FedGuCci *doesn't require additional communication costs compared with FedAvg*. FedGuCci uses historical global models, which are communicated in previous rounds and stored at the clients. Instead, FedGuCci may require additional storage at the clients for historical global models when $N > 1$, but the storage is lightweight and acceptable. For computation, in Table 8, we will show that FedGuCci is more efficient than the baselines given a computation budget.

## 4.2 FEDGUCCI+: ALIGNING LOCAL LOSS LANDSCAPES

In the study of LMC, different modes are trained on the ***same*** dataset but with different random seeds or initializations (Entezari et al., 2022). However, in FL, clients have ***heterogeneous*** data, and it is found that data heterogeneity of clients will cause different curvatures of local loss landscapes (Zhou et al., 2023), making the connectivity worse. Therefore, aligning local loss landscapes is essential for better performances of the connectivity loss. In this subsection, we incorporate previous techniques in FedGuCci to align local loss landscapes and propose FedGuCci+.

Table 3: **Results of pretrained language models on natural language processing (GLUE benchmark).**

| Methods/Tasks | SST-2 | MRPC | CoLA | QNLI | RTE | STS-B | AVG |
|---|---|---|---|---|---|---|---|
| Local | 92.55±0.19 | 78.38±0.37 | 47.98±1.01 | 84.66±0.10 | 55.69±1.03 | 87.11±0.36 | 75.40±0.51 |
| FedAvg | 92.79±0.24 | 84.17±0.38 | 53.86±0.70 | 84.52±0.14 | 68.63±1.53 | **88.61±0.34** | 78.76±0.56 |
| FedProx | 50.88±0.00 | 67.26±0.75 | 00.00±0.00 | 50.55±0.98 | 49.39±3.42 | 00.00±0.00 | 54.52±1.71 |
| FedDyn | 91.19±0.85 | 84.80±0.41 | **55.49±1.02** | 85.51±0.54 | 61.40±3.89 | 24.75±9.38 | 67.19±2.68 |
| SCAFFOLD | 92.75±0.12 | 84.11±0.65 | 54.28±0.31 | 84.73±0.16 | **69.24±2.76** | 88.31±0.31 | **78.90±0.72** |
| FedSAM | **92.79±0.14** | **84.81±0.08** | 53.25±0.43 | 82.13±0.34 | 68.14±2.09 | 87.71±0.42 | 78.14±0.58 |
| **FedGuCci** | **93.22±0.20** | **85.77±0.44** | 55.38±0.44 | 89.40±0.40 | 70.96±1.60 | 89.25±0.44 | 80.66±0.59 |

Table 4: **Results on different numbers of clients and participation ratios.** Non-IID hyper. is 1.0, and the dataset is CIFAR-10.

| $M$ | 100 | | 200 | |
|---|---|---|---|---|
| $\rho$ | 0.3 | 0.6 | 0.3 | 0.6 |
| Local | 27.91±0.24 | 27.53±0.10 | 23.39±0.18 | 23.20±0.22 |
| FedAvg | 63.98±0.84 | 63.41±0.55 | 61.37±0.79 | 61.15±1.01 |
| FedProx | 52.43±0.66 | 52.79±0.73 | 44.63±0.95 | 44.96±0.78 |
| FedRoD | 61.15±0.05 | 60.30±0.02 | 58.01±0.92 | 57.63±1.44 |
| FedLC | 63.70±0.69 | 63.24±0.70 | 60.99±0.66 | 60.67±0.81 |
| FedSAM | 64.87±0.58 | 64.45±0.22 | 62.33±0.56 | 61.93±0.90 |
| **FedGuCci** | **65.02±0.41** | **64.54±0.41** | **62.37±0.83** | **62.13±0.63** |
| **FedGuCci+** | **65.34±0.21** | **65.50±0.35** | **63.29±0.71** | **63.93±0.81** |

Table 5: **Results of global models under pretrain-finetune vision models.** Non-IID hyper. is 10.

| Dataset | CIFAR-10 | | CIFAR-100 | |
|---|---|---|---|---|
| Models | ResNet-18 | ViT | ResNet-18 | ViT |
| Local | 65.33±0.35 | 87.04±0.43 | 31.01±0.34 | 64.38±0.47 |
| FedAvg | 74.89±0.16 | 96.16±0.19 | 45.24±0.57 | 83.61±0.69 |
| FedProx | 50.61±0.81 | 96.32±0.21 | 4.29±0.38 | 78.49±1.92 |
| FedRoD | 74.91±0.17 | 96.18±0.18 | 45.19±0.76 | 83.64±0.35 |
| FedLC | 74.94±0.13 | 96.21±0.17 | 45.18±0.65 | 83.38±0.64 |
| FedSAM | 74.79±0.49 | 96.27±0.01 | 45.05±0.44 | 83.13±0.82 |
| **FedGuCci** | **75.22±0.12** | **96.38±0.11** | **45.62±0.61** | **83.71±0.48** |
| **FedGuCci+** | **75.30±0.53** | **96.73±0.13** | **46.09±0.55** | **83.96±0.67** |

**Bias reduction.** In FL, class imbalance (a.k.a. label skew) is a main cause of data heterogeneity, and previous works propose logit calibration (Zhang et al., 2022), balanced softmax (Chen & Chao, 2022), and other techniques (Li et al., 2023b; Acar et al., 2020) for reducing the bias caused by class imbalance. Here, we introduce the logit calibration technique used in FedLC (Zhang et al., 2022) for bias reduction. The main idea of logit calibration is to add additional terms to the logits to balance the overall class distributions. From Figure 5 (b), it demonstrates that logit calibration and other bias reduction methods can align the landscapes by making the local objectives more consistent.

**Flatter minima.** Sharpness-aware minimization (Foret et al., 2021; Kwon et al., 2021) (SAM) find flatter minima to improve generalization. SAM has also been introduced in FL for better generalization (Caldarola et al., 2022; Qu et al., 2022). In our paper, we find SAM can be used to align local loss landscapes by making the landscapes flatter, so we also incorporate it in FedGuCci+. From Figure 5 (c), if the landscapes are flatter, the overlap regions between two clients will increase. Therefore, it will have more aligned landscapes. Also, for FedGuCci, SAM makes the connectivity loss to learn a cylinder connected with the anchor model instead of a line (Wen et al., 2023), improving connectivity robustness and generalization. FedGuCci+ incorporates logit calibration and SAM into FedGuCci,

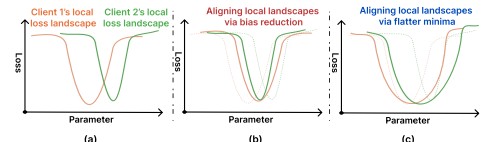

Figure 5: **Illustration of how FedGuCci+ aligns the local loss landscapes. (a):** Vanilla FedGuCci. Due to data heterogeneity, clients have different local loss landscapes. **(b):** FedGuCci+Bias Reduction. Introducing logit calibration or other FL bias reduction techniques can align the learning objectives. **(c):** FedGuCci+Flatter Minima. Introducing sharpness-aware minimization can make the landscapes flatter, and as a result, the overlapping regions increase.

achieving better generalization. We note that FedGucCci+ is a showcase of how FedGucCci is compatible with other existing techniques for better results, and more techniques can be integrated.

## 5 EXPERIMENTS

In this section, we conduct extensive experiments to validate how FedGuCci and FedGuCci+ improve the generalization of FL under various settings and datasets.

### 5.1 SETTINGS

**Datasets and models.** Following previous works (Li et al., 2023b; Lin et al., 2020; Li et al., 2023a), we use 4 vision datasets to conduct experiments: Fashion-MNIST (Xiao et al., 2017), CIFAR-10

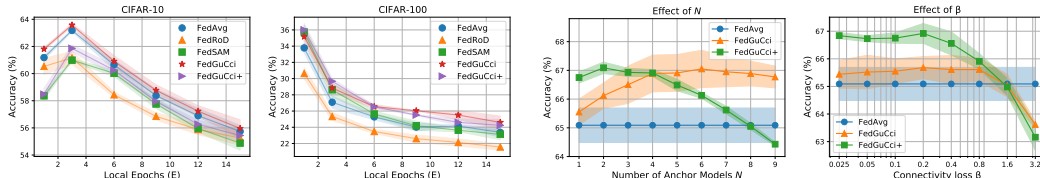

Figure 6: **Results under different epochs $E$.** $M = 60$ for CIFAR-10, and $M = 20$ for CIFAR-100. $T$ is 200 for both datasets. The non-IID hyper. is 0.4.

Figure 7: **Sensitivity analysis for hyperparameters $N$ and $\beta$ for FedGuCci(+).** $M = 60$ and non-IID hyperparameter is 0.4.

(Krizhevsky et al., 2009), CIFAR-100 (Krizhevsky et al., 2009), and Tiny-ImageNet (Le & Yang, 2015). Tiny-ImageNet is a subset of ImageNet (Deng et al., 2009) with 100k samples of 200 classes. We use different models for the datasets as follows: {Fashion-MNIST: VGG11 (Simonyan & Zisserman, 2015), CIFAR-10: SimpleCNN (Li et al., 2023a), CIFAR-100: ResNet20 (Li et al., 2018; He et al., 2016), Tiny-ImageNet: ResNet18 (He et al., 2016)}. We also conduct experiments of pretrained language models on 6 datasets are from GLUE (Wang et al., 2019), and the model is RoBERTa-base (Liu et al., 2019). For the detailed settings, please refer to Appendix A.

**Compared methods.** We take the most relevant and the most state-of-the-art FL algorithms as the baselines. *(1) FedAvg* (McMahan et al., 2017) with vanilla local training, a simple but strong baseline; *(2) FedProx* (Li et al., 2020a), which uses the current round's global model as local regularization term; *(3) FedDyn* (Acar et al., 2020), FL based on dynamic regularization; *(4) SCAF-FOLD* (Karimireddy et al., 2020), using control variates for variance reduction; *(5) MOON* (Li et al., 2021) with model-contrastive learning; *(6) FedRoD* (Chen & Chao, 2022), generalization through decoupling and balanced softmax loss; *(7) FedLC* (Zhang et al., 2022), FL with logit calibration for bias reduction; *(8) FedSAM* (Qu et al., 2022; Caldarola et al., 2022), incorporating sharpness-aware minimization into FL.

**Client Settings.** We adopt the Dirichlet sampling to craft IID and heterogeneous data for clients, which is widely used in FL literature (Lin et al., 2020; Chen & Chao, 2022; Li et al., 2023b). It considers a class-imbalanced data heterogeneity, controlled by non-IID hyperparameter, and smaller value refers to more heterogeneous data of clients. We vary the hyperparameter $\in \{100, 10, 1.0, 0.5, 0.4, 0.1\}$ with a spectrum from IID to non-IID (heterogeneous). The hyperparameters are shown in the captions or in Appendix A. Except from Table 4, we use full client participation.

**Evaluation and implementation.** We test the generalization performance, which is validated on the balanced testset after the global model is generated on the server. For all the experiments, we conduct three trials for each setting and present the mean accuracy and the standard deviation in the tables. More implementation details, e.g., hyperparameters, in Appendix A.

## 5.2 MAIN RESULTS

**Results under various datasets and models.** In Table 2, our methods can reach state-of-the-art results across four datasets under both IID ($\alpha = 100$) and heterogeneous ($\alpha = 0.5$) settings[2]. Generally, FedGuCci can reach the best performances over current FL methods, and FedGuCci+ can strengthen FedGuCci in most cases. Also, the performance gains of our approaches are more dominant under more complicated datasets, like Tiny-ImageNet. While FedSAM stands as the most robust baseline for generalization, our connectivity loss not only yields better results but is also compatible with it (FedGuCci+).

**Results on different $M$ and $\rho$.** We conduct experiments by varying the number of clients $M$ and participation ratios of clients $\rho$ in Table 4. It demonstrates that FedGuCci and FedGuCci+ can also excel when the number of clients is large and partial participation exists, indicating their great potential under cross-device settings (Charles et al., 2021).

**Results of different local epochs $E$.** In Figure 6, FedGuCci is consistently leading under different $E$, while FedGuCci+ is not robust on CIFAR-10. For CIFAR-100, FedGuCci has a more obvious advantage when $E$ is large, and this is rationale since the connectivity and model drift issues are more severe under large local updates.

---

[2]It's important to mention that certain methods might fail in specific settings, exhibiting accuracy levels close to random guessing, e.g., FedProx in Fashion-MNIST.

### 5.3 EXPERIMENTS UNDER PRETRAINED MODELS

We conduct experiments under pretrain-finetune paradigm for both vision and language tasks.

**Results under pretrained language models.** We use 6 datasets from GLUE (Wang et al., 2019) benchmark for finetuning pretrained language models. For each dataset, we randomly split the data into several clients and conduct finetuning using low-rank adaption (LoRA), and the pretrained model is RoBERTa-base (Liu et al., 2019). It is notable that some language tasks are not classifications, so FedRoD, FedLC, and FedGuCci+, which rely on classification loss, are not applicable. The results are in Table 3, where our FedGuCci reaches promising performances over existing methods. It is observed that some methods that are superior in Table 2 have worse performances in pretrained language models, e.g., FedDyn, while our FedGuCci keeps steady advantages.

**Results under pretrained vision models.** We conduct experiments under pretrained vision models, namely, ResNet18 (He et al., 2016) pretrained on ImageNet (Deng et al., 2009) and Vision Transformer (ViT-B/32) (Dosovitskiy et al., 2021) pretrained on CLIP (Radford et al., 2021). Table 5 presents the finetuning results of FL methods on CIFAR-10 and CIFAR-100. It seems that FedAvg is a strong baseline when it comes to pretrained vision backbones, especially for the ViT. However, it is illustrated that FedGuCci is also improving generalization over FedAvg.

In this subsection, we showcase the applicability of FedGuCci under the pretrain-finetune paradigm, and it reveals FedGuCci's great potential in collaboratively finetuning foundation models, such as large language models (Radford et al., 2018; Touvron et al., 2023).

### 5.4 FURTHER ANALYSES AND ABLATION STUDIES

**Sensitivity analyses of hyperparameters.** As illustrated in Figure 7, we vary the FedGuCci(+)'s hyperparameters $N$ and $\beta$ of Equation 11 and Equation 12. It reveals that FedGuCci and FedGuCci+ have a wide range of effective hyperparameters, outperforming FedAvg. We find FedGuCci+ is more sensitive than FedGuCci, that high $N$ and $\beta$ may degrade the performances. For $\beta$, there may exist an optimization-connectivity tradeoff at the clients. If $\beta$ is too high, the connectivity loss may hurt the local optimization steps, causing generalization declines of local models, further detrimental to the fused global model.

We conduct sensitivity analyses of FedGuCci(+)'s hyperparameters and their ablation study.

**Ablation study.** Table 6 shows that FedGuCci already has obvious generalization gains over FedAvg; further, SAM and the bias reduction method (logit calibration) can reach higher generalization on FedGuCci. SAM has a more dominant improvement on FedGuCci. We note that FedGuCci is general and flexible and may be compatible with more existing FL algorithms (Sun et al., 2023; Dai et al., 2023), and FedGuCci+ is just one showcase.

Table 6: **Ablation study of FedGuCci+.** $M = 50$, non-IID: 1.0.

| Methods/Datasets | CIFAR-10 | CIFAR-100 |
|---|---|---|
| FedAvg | $64.14_{\pm0.38}$ | $20.81_{\pm0.52}$ |
| FedGuCci | $65.45_{\pm0.19}$ | $22.74_{\pm0.42}$ |
| FedGuCci + only logit calibration | $65.51_{\pm0.15}$ | $22.99_{\pm0.58}$ |
| FedGuCci + only SAM | $\mathbf{65.93}_{\pm0.38}$ | $\mathbf{25.81}_{\pm1.02}$ |
| FedGuCci+ (with both) | $\mathbf{66.05}_{\pm0.35}$ | $\mathbf{25.97}_{\pm0.49}$ |

**Computation analysis.** In Table 8, we compare the computation costs of methods in terms of reaching a targeted accuracy. It can be seen that FedGuCci requires less computation to reach the target accuracies than the baselines, e.g., FedRoD, MOON, FedSAM, etc.

**More results:** Please refer to Appendix C for more results, including experiments under more heterogeneous data (Table 9 with non-IID hyper. 0.1 and 0.05), experiments under smaller participation ratios (Table 10), and so on.

## 6 CONCLUSION

In this paper, we study the transitivity of linear mode connectivity (LMC) and use this property to improve the generalization of federated learning (FL). We first empirically and theoretically verify the transitivity of LMC between two models by leveraging a fixed anchor model, and we extend it to group connectivity among multiple models. Then, we propose FedGuCci and FedGuCci+ in FL. Extensive experiments demonstrate our proposed methods can improve the generalization of FL under various settings.

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

# Appendix

In this appendix, we provide the details omitted in the main paper and more analyses and discussions.

- Appendix A: details of experimental setups (cf. section 3 and section 5 of the main paper).
- Appendix B: detailed proofs of Lemma 3.3, Theorem 3.5, and Theorem 3.8 (cf. section 3 of the main paper).
- Appendix C: additional results and analyses (cf. section 3 and section 5 of the main paper).
- Appendix D: more discussions about the related works (cf. section 2 of the main paper).

## A  IMPLEMENTATION DETAILS

---

**Algorithm 1 FedGuCci**: Federated Learning with Improved Group Connectivity

---

**Input**: $M$ clients, communication round $T$, local epoch $E$, participation ratio $\rho = \frac{K}{M}$; number of anchor models $N$; initial global model $\mathbf{w}_g^1$;

**Output**: final global model $\mathbf{w}_g^T$;

1: **for** each round $t = 1, \ldots, T$ **do**
2:     # Client updates
3:     **for** each client $i, i \in [M]$ **in parallel do**
4:         Set local model $\mathbf{w}_i^t \leftarrow \mathbf{w}_g^t$;
5:         Replay $N$ historical global models as the anchor models $\mathbf{W}_{\text{anc}*}^t$ by Equation 11;
6:         Compute $E$ epochs of client local training with connectivity loss by Equation 12;
7:     **end for**
8:     # Server updates
9:     The server samples a set $\mathcal{S}^t$ of $K$ clients and receive their models $\{\mathbf{w}_i^t\}_{i \in \mathcal{S}^t}$;
10:     The server obtains the global model $\mathbf{w}_g^{t+1}$ via aggregation by Equation 2;
11: **end for**
12: Obtain the final global model $\mathbf{w}_g^T$.

---

In this section, we present the implementation details omitted from the main paper.

### A.1  IMPLEMENTATION ENVIRONMENT

All experiments were conducted on Intel Xeon Silver 4108 CPU, and NVIDIA Tesla V100 GPU with 32GB of graphics memory, using Python 3.9.18 and PyTorch 2.1.0.

### A.2  DATASETS

**CIFAR-10** (Krizhevsky et al., 2009) consists of 60,000 32x32 color images, evenly distributed among 10 different classes, including airplanes, automobiles, birds, cats, etc., each represented by 6,000 images. The dataset is split into 50,000 training images and 10,000 test images. **FashionMNIST** (Xiao et al., 2017) is designed as an advanced replacement for the MNIST dataset, suitable for benchmarking machine learning models. It comprises 70,000 images divided into 60,000 training samples and 10,000 test samples. Each image is a 28x28 grayscale representation of fashion items from 10 different classes, such as shirts, trousers, sneakers, etc. The **CIFAR-100** dataset (Krizhevsky et al., 2009) is similar to CIFAR-10 but more challenging, containing 100 different classes grouped into 20 superclasses. It includes 60,000 32x32 color images, with 600 images per class, divided into 50,000 training images and 10,000 test images. This dataset is primarily used for developing and evaluating more sophisticated image classification models. **TinyImageNet** TinyImageNet is a reduced-scale version of the renowned ImageNet dataset, which comprises a total of 200 classes. The dataset is structured into training, validation, and test sets, with 200,000 training images, 20,000 validation images, and 20,000 test images. The **GLUE** benchmark is a compilation of 9 datasets for evaluating natural language understanding systems. Tasks are framed as either single-sentence classification or sentence-pair classification tasks. GLUE includes MNLI (inference, (Williams et al., 2017)), MRPC (paraphrase detection, (Socher et al., 2013)), MRPC (paraphrase detection, (Dolan & Brockett, 2005)),

CoLA (linguistic acceptability, (Warstadt et al., 2019)), QNLI (inference, (Rajpurkar et al., 2018)), QQP (question-answering), RTE (inference), WNLI (inference), and STS-B (textual similarity, (Cer et al., 2017)). Due to high computation costs, we only used SST2, MRPC, CoLA, QNLI, RTE, and STS-B for evaluation. For the replication in Table 3, we report results on the development sets after fine-tuning the pretrained models on the corresponding single-task training data. Our fine-tuning approach is LoRA(Hu et al., 2021).

### A.3 MODELS

**SimpleCNN.** The simple CNN for CIFAR-10 is a convolutional neural network model with ReLU activations, consisting of 3 convolutional layers followed by 2 fully connected layers. The first convolutional layer has a size of (3, 32, 3), followed by a max-pooling layer of size (2, 2). The second and third convolutional layers have sizes of (32, 64, 3) and (64, 64, 3), respectively. The last two fully connected layers have sizes of (6444, 64) and (64, num_classes), respectively.

**ResNets.** We followed the model architectures used in (Li et al., 2018). The number in the model names indicates the number of layers in the models, whereas a larger number indicates a deeper network. We used ResNet18 and ResNet20 for CIFAR-10 and CIFAR-100, respectively. Notably, to mitigate abnormal effects introduced by batch normalization layers (Li et al., 2020b; Lin et al., 2020), followed by (Adilova et al., 2023), we removed all batch normalization layers from the ResNets.

**VGG.** VGG (Simonyan & Zisserman, 2015) is a convolutional neural network (CNN) architecture that gained prominence in the field of computer vision. Among its variants, we used VGG11.

**RoBERTa.** RoBERTa is a natural language processing (NLP) model that builds upon the foundation laid by BERT, which was introduced by (Liu et al., 2019) to address some limitations and improve the performance of BERT on various NLP tasks. It comes in various sizes, and we used RoBERTa-base considering to high computational costs.

**ViT.** ViT (Dosovitskiy et al., 2020) is a deep learning model for visual tasks that adopts the Transformer structure proposed in NLP. ViT divides a picture into several patches, treats the patch as a word, and then uses a self-attention mechanism to capture the relationship between patches. When ViT is pre-trained with a large amount of data, it will perform particularly well on downstream tasks.

### A.4 RANDOMNESS

In all experiments, we conducted each experiment three times with different random seeds and reported the averaged results along with standard deviations.

We ensured consistency by setting torch, numpy, and random functions with the same random seed, thereby making the data partitions and other settings identical. To ensure all algorithms started with the same initial model, we saved an initial model for each architecture and loaded it at the beginning of each experiment. Additionally, for experiments involving partial participation, the selection of participating clients in each round significantly influenced the model's performance. To maintain fairness, we saved the sequences of participating clients in each round and loaded these sequences for all experiments. This procedure guaranteed that, given a random seed and participation ratio, every algorithm had the same set of sampled clients in each round.

### A.5 EVALUATION

**CIFAR-10, CIFAR-100, FashionMNIST and Tiny-ImageNet.** We evaluate the global model performance on the test dataset of each dataset. The test dataset is mostly class-balanced and can reflect the global learning objective of a federated learning system. Therefore, the performance of the model on the test set can indicate the generalization performance of global models (Li et al., 2023a; Lin et al., 2020). In each experiment, we take the average test accuracy of the last 5 rounds as the final test accuracy.

**GLUE.** For GLUE, we used the validation dataset for evaluation. Following by (Hu et al., 2021), we chose the best accuracy as the final test accuracy.

## A.6 Hyperparameter

Table 2: For Fashion-MNIST, $T$ is 400, batch size is 64 and learning rate is 0.08. For CIFAR-10, $T$ is 150, batch size is 64 and learning rate is 0.04. For CIFAR-100, $T$ is 200, batch size is 64 and learning rate is 0.03. For Tiny-ImageNet, learning rate is 0.01 and $T$ is 50. Optimzier is ADAM for Fashion-MNIST and others are SGD.

Table 3: Optimizer is Adam for all datasets. For CoLA and STSB, $T$ is 25, batch size is 16 and learning rate is 2e-5. For SST-2, $T$ is 50, batch size is 16, and learning rate is 2e-6. For QNLI, $T$ is 20, batch size is 32 and learning rate is 2e-6. For RTE and MRPC, $T$ is 80, batch size is 16 and learning rate is 2e-5.

Table 4: $T$ is 150, E is 3, batch size is 64 and learning rate is 0.04.

Table 5: ResNet-18 and MobileViT are pretrained on ImageNet. E is 3 for both models. For ViT, $T$ is 15, batch size is 16 and learning rate is 0.001. For ResNet, $T$ is 50, batch size is 64 and learning rate 1e-4.

Table 6: For CIFAR-10, $T$ is 150, batch size is 64 and learning rate is 0.04. For CIFAR-100, T is 200, batch size is 64, and learning rate is 0.03.

Figure 6: $M = 60$ for CIFAR-10, and $M = 20$ for CIFAR-100. $T$ is 200 for both datasets. Learning rate is 0.03 for CIFAR-10, and 0.04 for CIFAR-100.

Figure 7: $T$ is 150, $E$ is 3, $M$ is 60, learning rate is 0.02, and batch size is 64.

## B Proof

In this section, we give the proofs of the lemma and theorem in section 3.

**Lemma B.1** *(Lemma 3.3) Set the uniform and bounded domain for network* $\mathbf{w}$ *as* $\mathcal{E}_\epsilon = \{\mathbf{w} \in \Omega | \mathcal{L}(\mathbf{w}) < \epsilon\}$. *Define a random event* $D_\epsilon(\mathbf{w}^*_{anc})$ *as* $D_\epsilon(\mathbf{w}^*_{anc}) = \{\exists \mathbf{w} \in \mathcal{E}_\epsilon | \forall \alpha \in [0,1], \mathcal{L}(\alpha \mathbf{w}^*_{anc} + (1-\alpha)\mathbf{w}) \leq \epsilon\}$. *Consider an anchor model* $\mathbf{w}^*_{anc}$ *and an arbitrary network* $\mathbf{w}$ *and for* $\epsilon > 0$. *Then for* $\|\mathbf{w} - \mathbf{w}^*_{anc}\|_\infty \leq \frac{d}{2}$,

$$P(D_\epsilon(\mathbf{w}^*_{anc})) \leq (\frac{d_\epsilon}{d})^S, \tag{13}$$

*where* $d_\epsilon = |\mathcal{E}_\epsilon|^{\frac{1}{S}}$ *represents the average diameter of region* $\mathcal{E}_\epsilon$, $S$ *represents the number of parameters of the neural network and the equality holds if and only if* $\mathcal{E}_\epsilon \subset \{\mathbf{w} | \|\mathbf{w} - \mathbf{w}^*_{anc}\|_\infty \leq d\}$ *is a star domain centered at* $\mathbf{w}^*_{anc}$. *Thus, when* $P(D_\epsilon(\mathbf{w}^*_{anc}))) > 1 - \delta$, *it holds* $d < \frac{d_\epsilon}{(1-\delta)^{\frac{1}{S}}}$.

*Proof:* In the following proof, we denote the region as $\mathcal{V}_d = \{\mathbf{w} | \|\mathbf{w} - \mathbf{w}^*_{anc}\|_\infty \leq \frac{d}{2}\}$ with volume $|\mathcal{V}_d| = d^S$ and denote the segment between $\mathbf{w}$ and $\mathbf{w}^*_{anc}$ as $l(\mathbf{w}^*_{anc}, \mathbf{w}) = \{\alpha \mathbf{w}^*_{anc} + (1-\alpha)\mathbf{w}, \alpha \in [0,1]\}$.

First we prove if $\mathcal{E}_\epsilon \subset \mathcal{V}_d$ is a star domain centered at $\mathbf{w}^*_{anc}$, $P(D_\epsilon(\mathbf{w}^*_{anc})) = \frac{|\mathcal{E}_\epsilon|}{d^S}$. Select a parameter point $\mathbf{w}_0$ in $\mathcal{V}_d$ arbitrarily. If $\mathbf{w}_0 \in \mathcal{E}_\epsilon$, then because $\mathcal{E}_\epsilon$ is a star domain centered at $\mathbf{w}^*_{anc}$, $l(\mathbf{w}^*_{anc}, \mathbf{w}) \subset \mathcal{E}_\epsilon$ and thus $\mathbf{w}_0 \in D_\epsilon(\mathbf{w}^*_{anc})$. If $\mathbf{w}_0 \notin \mathcal{E}_\epsilon$, then $\mathbf{w}_0 \notin D_\epsilon(\mathbf{w}^*_{anc})$ by the definition of $D_\epsilon(\mathbf{w}^*_{anc})$. Therefore, $\mathcal{E}_\epsilon = D_\epsilon(\mathbf{w}^*_{anc})$ and we have $P(D_\epsilon(\mathbf{w}^*_{anc})) = P(\mathcal{E}_\epsilon) = \frac{|\mathcal{E}_\epsilon|}{|\mathcal{V}_d|} = \frac{|\mathcal{E}_\epsilon|}{d^S}$.

The next step we prove that if $\mathcal{E}_\epsilon \not\subset \mathcal{V}_d$, or $\mathcal{E}_\epsilon$ is not a star domain centered at $\mathbf{w}^*_{anc}$, then $P(D_\epsilon(\mathbf{w}^*_{anc})) < \frac{|\mathcal{E}_\epsilon|}{d^S}$.

If $\mathcal{E}_\epsilon \not\subset \mathcal{V}_d$, then $|D_\epsilon(\mathbf{w}^*_{anc})| \leq |\mathcal{E}_\epsilon \cap \mathcal{V}_d| < |\mathcal{E}_\epsilon|$ and $P(D_\epsilon(\mathbf{w}^*_{anc})) = \frac{|D_\epsilon(\mathbf{w}^*_{anc})|}{|\mathcal{V}_d|} < \frac{|\mathcal{E}_\epsilon|}{|\mathcal{V}_d|}$. Here, the first inequality $|D_\epsilon(\mathbf{w}^*_{anc})| \leq |\mathcal{E}_\epsilon \cap \mathcal{V}_d|$ holds, because $D_\epsilon(\mathbf{w}^*_{anc}) \subset \mathcal{E}_\epsilon \cap \mathcal{V}_d$ and the second inequality $|\mathcal{E}_\epsilon \cap \mathcal{V}_d| < |\mathcal{E}_\epsilon|$ holds, because $\exists \mathbf{w}_0 \in \mathcal{E}_\epsilon/\mathcal{V}_d, \epsilon_0 > 0$ st. $\{\mathbf{w} | \|\mathbf{w} - \mathbf{w}_0\| < \epsilon_0\} \subset \Omega/\mathcal{V}_d \cap \mathcal{E}_\epsilon$ for $\Omega/\mathcal{V}_d$ and $\mathcal{E}_\epsilon$ are open sets and $|\mathcal{E}_\epsilon \cap \mathcal{V}_d| \leq |\mathcal{E}_\epsilon| - |\{\mathbf{w} | \|\mathbf{w} - \mathbf{w}_0\| < \epsilon_0\}| < |\mathcal{E}_\epsilon|$.

If $\mathcal{E}_\epsilon$ is not a star domain centered at $\mathbf{w}^*_{anc}$, then there exists $\mathbf{w}_0 \in \mathcal{E}_\epsilon$ such that $l(\mathbf{w}^*_{anc}, \mathbf{w}_0) \not\subset \mathcal{E}_\epsilon$. Then $\exists \alpha_1 \in (0,1)$ st. $\mathbf{w}_1 \overset{\Delta}{=} \alpha_1 \mathbf{w}^*_{anc} + (1-\alpha_1)\mathbf{w}_0$ satisfies $\mathcal{L}(\mathbf{w}_1) > \epsilon$. For $\mathcal{L}(\cdot)$ is smooth, there exists

$\epsilon_1 > 0$ st. $\forall \mathbf{w} \in U_{\epsilon_1}(\mathbf{w}_1) \triangleq \{\mathbf{w} | \|\mathbf{w}_1 - \mathbf{w}\|_2 < \epsilon_1\}$, $\mathcal{L}(\mathbf{w}) \geq \epsilon + \frac{\mathcal{L}(\mathbf{w}_1) - \epsilon}{2} > \epsilon$. Then for $\mathcal{E}_\epsilon$ is an open set, choose $\epsilon_2 < \epsilon_1$ st. $U_{\epsilon_2}(\mathbf{w}_0) \subset \mathcal{E}_\epsilon$. $\forall \mathbf{w}_2 \in U_{\epsilon_2}(\mathbf{w}_0)$, $\mathbf{w}_3 = \alpha_1 \mathbf{w}_{anc}^* + (1 - \alpha_1) \mathbf{w}_2$ satisfies $\|\mathbf{w}_3 - \mathbf{w}_1\|_2 = (1 - \alpha_1)\|\mathbf{w}_0 - \mathbf{w}_2\|_2 < (1 - \alpha_1)\epsilon_2 < \epsilon_1$. Thus $\mathbf{w}_3 \in U_{\epsilon_1}(\mathbf{w}_1)$, which leads to $\mathcal{L}(\mathbf{w}_3) > \epsilon$. Therefore, $U_{\epsilon_2}(\mathbf{w}_0) \cap D_\epsilon(\mathbf{w}_{anc}^*) = \emptyset$ and $P(D_\epsilon(\mathbf{w}_{anc}^*)) = \frac{|D_\epsilon(\mathbf{w}_{anc}^*)|}{d^S} \leq \frac{|\mathcal{E}_\epsilon| - |U_{\epsilon_2}(\mathbf{w}_0)|}{d^S} < \frac{|\mathcal{E}_\epsilon|}{d^S}$. $\qquad\square$

**Theorem B.2** (*Theorem 3.5*) *We define a two-layer neural network with ReLU activation, and the function is $f_{\mathbf{v}, \mathbf{U}}(\mathbf{x}) = \mathbf{v}^\top \sigma(\mathbf{U}\mathbf{x})$ where $\sigma(\cdot)$ is the ReLU activation function. $\mathbf{v} \in \mathbb{R}^h$ and $\mathbf{U} \in \mathbb{R}^{h \times l}$ are parameters[3] and $\mathbf{x} \in \mathbb{R}^l$ is the input which is taken from $\mathbb{X} = \{\mathbf{x} \in \mathbb{R}^l | \|\mathbf{x}\|_2 < b\}$ uniformly. Denote the deterministic anchor model as $\mathbf{w}_{anc}^* = \{\mathbf{U}_{anc}^*, \mathbf{v}_{anc}^*\}$, with $\|\mathbf{v}_{anc}^*\|_2 < d_{anc}$ and consider two different networks $\mathbf{w}_1, \mathbf{w}_2$ parameterized with $\{\mathbf{U}_1, \mathbf{v}_1\}$ and $\{\mathbf{U}_2, \mathbf{v}_2\}$ respectively. Each element of $\mathbf{U}_1$ and $\mathbf{U}_2$, $\mathbf{v}_1$ and $\mathbf{v}_2$ is sampled from a uniform distribution centered at $\mathbf{U}_{anc}^*$ and $\mathbf{v}_{anc}$ with an interval length of $d$. If with probability $1 - \delta$, $\sup_\alpha \mathcal{L}(\alpha \mathbf{w}_{anc}^* + (1 - \alpha)\mathbf{w}_1) < \epsilon$ and $\sup_\alpha \mathcal{L}(\alpha \mathbf{w}_{anc}^* + (1 - \alpha)\mathbf{w}_2) < \epsilon$, then with probability $1 - \delta$, it has,*

$$B_{loss}(\mathbf{w}_1, \mathbf{w}_2) \leq \frac{\sqrt{2h}b}{2(1 - \delta)^{\frac{2}{hl+h}}} d_\epsilon(d_\epsilon + d_{anc}) \log(12h/\delta), \tag{14}$$

*where $B_{loss}(\mathbf{w}_1, \mathbf{w}_2)$ is the loss barrier as Equation 3.*

*Proof:* Let's first define $g_\alpha(\mathbf{x}) = (\alpha \mathbf{U}_1 + (1 - \alpha)\mathbf{U}_2)\mathbf{x}$ and $z_{\mathbf{x}}(\alpha) = (\alpha \mathbf{v}_1 + (1 - \alpha)\mathbf{v}_2)^\top \sigma((\alpha \mathbf{U}_1 + (1 - \alpha)\mathbf{U}_2)\mathbf{x}) - \alpha \mathbf{v}_1^\top \sigma(\mathbf{U}_1 \mathbf{x}) - (1 - \alpha)\mathbf{v}_2^\top \sigma(\mathbf{U}_2 \mathbf{x})$, $\alpha \in [0, 1]$. Then we can express $z_{\mathbf{x}}(\alpha)$ as:

$$z_{\mathbf{x}}(\alpha) = (\alpha \mathbf{v}_1 + (1 - \alpha)\mathbf{v}_2)^\top \sigma(g_\alpha(\mathbf{x})) - \alpha \mathbf{v}_1^\top \sigma(\mathbf{U}_1 \mathbf{x}) - (1 - \alpha)\mathbf{v}_2^\top \sigma(\mathbf{U}_2 \mathbf{x}). \tag{15}$$

For each element of $\mathbf{U}_1$ and $\mathbf{U}_2$, $\mathbf{v}_1$ and $\mathbf{v}_2$ is sampled from a uniform distribution centered at $\mathbf{U}_{anc}^*$ and $\mathbf{v}_{anc}^*$ with an interval length of $d$, $\mathbf{U}_1, \mathbf{U}_2, \mathbf{v}_1$ and $\mathbf{v}_2$ can be represented as $\mathbf{U}_1 = \mathbf{U}_{anc}^* + \tilde{\mathbf{U}}_1$, $\mathbf{U}_2 = \mathbf{U}_{anc}^* + \tilde{\mathbf{U}}_2$, $\mathbf{v}_1 = \mathbf{v}_{anc}^* + \tilde{\mathbf{v}}_1$ and $\mathbf{v}_2 = \mathbf{v}_{anc}^* + \tilde{\mathbf{v}}_2$ respectively, where each element of $\tilde{\mathbf{U}}_1, \tilde{\mathbf{U}}_2$, $\tilde{\mathbf{v}}_1$ and $\tilde{\mathbf{v}}_2$ follows distribution $U[-\frac{d}{2}, \frac{d}{2}]$. Using $\tilde{\mathbf{v}}_1$ and $\tilde{\mathbf{v}}_2$, $z_{\mathbf{x}}(\alpha)$ can be represented as

$$\begin{aligned}
z_{\mathbf{x}}(\alpha) &= (\alpha \mathbf{v}_1 + (1 - \alpha)\mathbf{v}_2)^\top \sigma(g_\alpha(\mathbf{x})) - \alpha \mathbf{v}_1^\top \sigma(\mathbf{U}_1 \mathbf{x}) - (1 - \alpha)\mathbf{v}_2^\top \sigma(\mathbf{U}_2 \mathbf{x}) \\
&= (\alpha \tilde{\mathbf{v}}_1 + (1 - \alpha)\tilde{\mathbf{v}}_2 + \mathbf{v}_{anc}^*)^\top \sigma(g_\alpha(\mathbf{x})) - \alpha(\tilde{\mathbf{v}}_1^\top + \mathbf{v}_{anc}^{*\top})\sigma(\mathbf{U}_1 \mathbf{x}) - (1 - \alpha)(\tilde{\mathbf{v}}_2^\top + \mathbf{v}_{anc}^{*\top})\sigma(\mathbf{U}_2 \mathbf{x}) \\
&= [(\alpha \tilde{\mathbf{v}}_1 + (1 - \alpha)\tilde{\mathbf{v}}_2)^\top \sigma(g_\alpha(\mathbf{x})) - \alpha \tilde{\mathbf{v}}_1^\top \sigma(\mathbf{U}_1 \mathbf{x}) - (1 - \alpha)\tilde{\mathbf{v}}_2^\top \sigma(\mathbf{U}_2 \mathbf{x})] \\
&\quad + \mathbf{v}_{anc}^{*\top}[\sigma(g_\alpha(\mathbf{x})) - \alpha \sigma(\mathbf{U}_1 \mathbf{x}) - (1 - \alpha)\sigma(\mathbf{U}_2 \mathbf{x})].
\end{aligned} \tag{16}$$

We also assume that the number of hidden neurons $h$ is sufficiently large for the convenience of analysis as (Entezari et al., 2022). In the following proof, we will make use of Hoeffding's inequality for sub-Gaussian distributions (especially, uniform distribution). Here, we state it for reference: Let $X_1, \ldots, X_n$ be $n$ independent random variables such that $X_i \sim U(-\frac{d}{2}, -\frac{d}{2})$. Then for any $\mathbf{a} = (a_1, \ldots, a_n) \in \mathbb{R}^n$, we have

$$\mathbb{P}\left[|\sum_{i=1}^n a_i X_i| > t\right] \leq 2 \exp\left(-\frac{2t^2}{d^2 \|a\|_2^2}\right).$$

To bound $z_{\mathbf{x}}(\alpha)$, we have

$$\begin{aligned}
|z_{\mathbf{x}}(\alpha)| &\leq |[(\alpha \tilde{\mathbf{v}}_1 + (1 - \alpha)\tilde{\mathbf{v}}_2)^\top \sigma(g_\alpha(\mathbf{x})) - \alpha \tilde{\mathbf{v}}_1^\top \sigma(\mathbf{U}_1 \mathbf{x}) - (1 - \alpha)\tilde{\mathbf{v}}_2^\top \sigma(\mathbf{U}_2 \mathbf{x})]| \\
&\quad + |\mathbf{v}_{anc}^{*\top}[\sigma(g_\alpha(\mathbf{x})) - \alpha \sigma(\mathbf{U}_1 \mathbf{x}) - (1 - \alpha)\sigma(\mathbf{U}_2 \mathbf{x})]| \\
&\leq \alpha |\tilde{\mathbf{v}}_1^\top(\sigma(g_\alpha(\mathbf{x})) - \sigma(\mathbf{U}_1 \mathbf{x}))| + (1 - \alpha)|\tilde{\mathbf{v}}_2^\top(\sigma(g_\alpha(\mathbf{x})) - \sigma(\mathbf{U}_2 \mathbf{x}))| \\
&\quad + \alpha |\mathbf{v}_{anc}^{*\top}(\sigma(g_\alpha(\mathbf{x})) - \sigma(\mathbf{U}_1 \mathbf{x}))| + (1 - \alpha)|\mathbf{v}_{anc}^{*\top}(\sigma(g_\alpha(\mathbf{x})) - \sigma(\mathbf{U}_2 \mathbf{x}))|.
\end{aligned} \tag{17}$$

Then we bound the first term and the third term, and the second term and the fourth term are bounded similarly due to symmetry. For the **concentration upper bound** of the first term of Equation 17, we

---

[3]For simplicity and without loss of generality, we omit the bias terms.

use the Hoeffding's inequality for elements of $\tilde{\mathbf{v}}_1$, with probability $1 - \frac{\delta}{k}$

$$\alpha \left| \tilde{\mathbf{v}}_1^\top \left[ (\sigma(g_\alpha(\mathbf{x})) - \sigma(\mathbf{U}_1\mathbf{x})) \right] \right| \leq \alpha d \sqrt{\frac{1}{2} \log(2k/\delta)} \| \sigma(g_\alpha(\mathbf{x})) - \sigma(\mathbf{U}_1\mathbf{x}) \|_2 \tag{18}$$

$$\leq \alpha d \sqrt{\frac{1}{2} \log(2k/\delta)} \| g_\alpha(\mathbf{x}) - \mathbf{U}_1\mathbf{x} \|_2 \tag{19}$$

$$= \alpha(1-\alpha) d \sqrt{\frac{1}{2} \log(2k/\delta)} \| (\mathbf{U}_2 - \mathbf{U}_1)\mathbf{x} \|_2. \tag{20}$$

Equation 19 is due to the fact that the ReLU activation function satisfies the Lipschitz continuous condition with constant 1. For the bound of the third term of Equation 17, we have

$$\alpha \left| \mathbf{v}_{\text{anc}}^{*\top} \left[ (\sigma(g_\alpha(\mathbf{x})) - \sigma(\mathbf{U}_1\mathbf{x})) \right] \right| \leq \alpha d_{\text{anc}} \| \sigma(g_\alpha(\mathbf{x})) - \sigma(\mathbf{U}_1\mathbf{x}) \|_2 \tag{21}$$

$$\leq \alpha d_{\text{anc}} \| g_\alpha(\mathbf{x}) - \mathbf{U}_1\mathbf{x} \|_2 \tag{22}$$

$$= \alpha(1-\alpha) d_{\text{anc}} \| (\mathbf{U}_2 - \mathbf{U}_1)\mathbf{x} \|_2. \tag{23}$$

Equation 22 is due to the fact that the ReLU activation function satisfies the Lipschitz continuous condition with constant 1. For the term $\| (\mathbf{U}_2 - \mathbf{U}_1)\mathbf{x} \|_2$ in Equation 20 and Equation 23, taking a union bound, with probability $1 - \frac{\delta}{k}$, we have

$$\| (\mathbf{U}_2 - \mathbf{U}_1)\mathbf{x} \|_2 \leq \sqrt{\sum_{i=1}^{h} | (\mathbf{U}_{B;i,:} - \mathbf{U}_{A;i,:})\mathbf{x} |^2} \tag{24}$$

$$= \sqrt{\sum_{i=1}^{h} | (\mathbf{U}_{B;i,:} - \mathbf{U}_{A;i,:})\mathbf{x} |^2} \tag{25}$$

$$\leq d \|\mathbf{x}\|_2 \sqrt{h \log(2hk/\delta)} \tag{26}$$

$$= db \sqrt{h \log(2hk/\delta)}. \tag{27}$$

Then take a union bound choosing $k = 6$ (because the union bound is taken for 6 equations, Equation 20 and Equation 27 for the first and the second terms in Equation 17 respectively, and Equation 27 for the third and the fourth terms in Equation 17 respectively.), with probability $1 - \delta$ we have

$$|z_{\mathbf{x}}(\alpha)| \leq \alpha \left| \tilde{\mathbf{v}}_1^\top (\sigma(g_\alpha(\mathbf{x})) - \sigma(\mathbf{U}_1\mathbf{x})) \right| + (1-\alpha) \left| \tilde{\mathbf{v}}_2^\top (\sigma(g_\alpha(\mathbf{x})) - \sigma(\mathbf{U}_2\mathbf{x})) \right| \tag{28}$$

$$+ \alpha \left| \mathbf{v}_{\text{anc}}^{*\top} (\sigma(g_\alpha(\mathbf{x})) - \sigma(\mathbf{U}_1\mathbf{x})) \right| + (1-\alpha) \left| \mathbf{v}_{\text{anc}}^{*\top} (\sigma(g_\alpha(\mathbf{x})) - \sigma(\mathbf{U}_2\mathbf{x})) \right| \tag{29}$$

$$\leq 2\alpha(1-\alpha) d \sqrt{\frac{1}{2} \log(12/\delta)} \cdot db \sqrt{h \log(12h/\delta)} + 2\alpha(1-\alpha) d_{\text{anc}} \cdot db \sqrt{h \log(12h/\delta)} \tag{30}$$

$$\leq 2\sqrt{2} \alpha(1-\alpha) \sqrt{h} b (d^2 + d d_{\text{anc}}) \log(12h/\delta) \tag{31}$$

$$\leq \frac{\sqrt{2}}{2} \sqrt{h} b (d^2 + d d_{\text{anc}}) \log(12h/\delta). \tag{32}$$

For $\sup_\alpha \mathcal{L}(\alpha \mathbf{w}_{\text{anc}}^* + (1-\alpha)\mathbf{w}) < \epsilon$ holds with probability $1-\delta$, by Lemma 3.3, we have $d < \frac{d_\epsilon}{(1-\delta)^{\frac{1}{S}}}$ with $S = hl + h$. Then $|z_{\mathbf{x}}(\alpha)|$ can be bounded as

$$|z_{\mathbf{x}}(\alpha)| \leq \frac{\sqrt{2h} b}{2(1-\delta)^{\frac{2}{hl+h}}} d_\epsilon (d_\epsilon + d_{\text{anc}}) \log(12h/\delta). \tag{33}$$

Now we turn to calculate the bound of the loss barrier $B_{loss}(\mathbf{w}_1, \mathbf{w}_2)$. For the loss function $L(\cdot, y)$ is convex and 1-Lipschitz, we have:

$$B_{loss}(\mathbf{w}_1, \mathbf{w}_2) = \mathbb{E}[L(f_{\alpha\mathbf{v}_1+(1-\alpha)\mathbf{v}_2, \alpha\mathbf{U}_1+(1-\alpha)\mathbf{U}_2}(\mathbf{x}), y) - \alpha L(f_{\mathbf{v}_1,\mathbf{U}_1}(\mathbf{x}), y) - (1-\alpha)L(f_{\mathbf{v}_2,\mathbf{U}_2}(\mathbf{x}), y)] \tag{34}$$

$$\leq \mathbb{E}[L(f_{\alpha\mathbf{v}_1+(1-\alpha)\mathbf{v}_2, \alpha\mathbf{U}_1+(1-\alpha)\mathbf{U}_2}(\mathbf{x}), y) - L(\alpha f_{\mathbf{v}_1,\mathbf{U}_1}(\mathbf{x}) + (1-\alpha)f_{\mathbf{v}_2,\mathbf{U}_2}(\mathbf{x}), y)] \tag{35}$$

$$\leq \mathbb{E}[|f_{\alpha\mathbf{v}_1+(1-\alpha)\mathbf{v}_2, \alpha\mathbf{U}_1+(1-\alpha)\mathbf{U}_2}(\mathbf{x}) - (\alpha f_{\mathbf{v}_1,\mathbf{U}_1}(\mathbf{x}) + (1-\alpha)f_{\mathbf{v}_2,\mathbf{U}_2}(\mathbf{x}))|], \tag{36}$$

where the expectation is with respect to the dataset. Equation 35 is due to the convexity of $L(\cdot, y)$, while Equation 36 is due to the assumption that $L(\cdot, y)$ is 1-Lipschitz. Then use the bound of $z_{\mathbf{x}}(\alpha)$, with probability $1 - \delta$, we have

$$B_{loss}(\mathbf{w}_1, \mathbf{w}_2) \leq \frac{\sqrt{2h}b}{2(1-\delta)^{\frac{2}{hl+h}}} d_\epsilon (d_\epsilon + d_{anc}) \log(12h/\delta). \tag{37}$$

$\square$

**Theorem B.3** *(Theorem 3.8) We define a two-layer neural network with ReLU activation, and the function is $f_{\mathbf{v},\mathbf{U}}(\mathbf{x}) = \mathbf{v}^\top \sigma(\mathbf{U}\mathbf{x})$ where $\sigma(\cdot)$ is the ReLU activation function. $\mathbf{v} \in \mathbb{R}^h$ and $\mathbf{U} \in \mathbb{R}^{h \times l}$ are parameters and $\mathbf{x} \in \mathbb{R}^l$ is the input which is taken from $\mathbb{X} = \{\mathbf{x} \in \mathbb{R}^l | \|\mathbf{x}\|_2 < b\}$ uniformly. Denote the deterministic anchor model as $\mathbf{w}_{anc}^* = \{\mathbf{U}_{anc}^*, \mathbf{v}_{anc}^*\}$, with $\|\mathbf{v}_{anc}^*\|_2 < d_{anc}$ and consider $K$ different networks $\mathbf{w}_i$ parameterized with $\{\mathbf{U}_i, \mathbf{v}_i\}$ located on $K$ clients respectively. Each element of $\mathbf{U}_i$ and $\mathbf{v}_i$ is sampled from a uniform distribution centered at $\mathbf{U}_{anc}^*$ and $\mathbf{v}_{anc}^*$ with an interval length of $d$. If with probability $1 - \delta$, $\sup_\alpha \mathcal{L}_i(\alpha\mathbf{w}_{anc}^* + (1-\alpha)\mathbf{w}_i) < \epsilon$, then with probability $1 - \delta$, it has,*

$$B_{loss}(\{\mathbf{w}_i\}_{i=1}^K) \leq \tag{38}$$

$$\frac{\sqrt{2h}b}{2(1-\delta)^{\frac{2}{hl+h}}} d_{\epsilon+\Gamma} (d_{\epsilon+\Gamma} + d_{anc}) \log(4hK^2/\delta).$$

*Proof:* Similar to Theorem 3.5, we first define $g(\mathbf{x}) = (\frac{1}{K}\sum_{i=1}^K \mathbf{U}_i)\mathbf{x}$ and $z(\mathbf{x}) = (\frac{1}{K}\sum_{i=1}^K \mathbf{v}_i)^\top \sigma((\frac{1}{K}\sum_{i=1}^K \mathbf{U}_i)\mathbf{x}) - \frac{1}{K}\sum_{i=1}^K \mathbf{v}_i\sigma(\mathbf{U}_i\mathbf{x})$. Then we can express $z(\mathbf{x})$ as:

$$z(\mathbf{x}) = (\frac{1}{K}\sum_{i=1}^K \mathbf{v}_i)^\top \sigma(g(\mathbf{x})) - \frac{1}{K}\sum_{i=1}^K \mathbf{v}_i^\top \sigma(\mathbf{U}_i\mathbf{x}). \tag{39}$$

For each element of $\mathbf{U}_i$ and $\mathbf{v}_i$ is sampled from a uniform distribution centered at $\mathbf{U}_{anc}^*$ and $\mathbf{v}_{anc}^*$ with an interval length of $d$, $\mathbf{U}_i$ and $\mathbf{v}_i$ can be represented as $\mathbf{U}_i = \mathbf{U}_{anc}^* + \tilde{\mathbf{U}}_i$ and $\mathbf{v}_i = \mathbf{v}_{anc}^* + \tilde{\mathbf{v}}_i$ respectively, where each element of $\tilde{\mathbf{U}}_i$ and $\tilde{\mathbf{v}}_i$ follows distribution $U[-\frac{d}{2}, \frac{d}{2}]$. Using $\tilde{\mathbf{v}}_i$, $z_{\mathbf{x}}(\alpha)$ can be represented as

$$z(\mathbf{x}) = (\frac{1}{K}\sum_{i=1}^K \mathbf{v}_i)^\top \sigma(g(\mathbf{x})) - \frac{1}{K}\sum_{i=1}^K \mathbf{v}_i^\top \sigma(\mathbf{U}_i\mathbf{x}) \tag{40}$$

$$= (\mathbf{v}_{anc}^* + \frac{1}{K}\sum_{i=1}^K \tilde{\mathbf{v}}_i)^\top \sigma(g(\mathbf{x})) - \frac{1}{K}\sum_{i=1}^K (\mathbf{v}_{anc}^* + \tilde{\mathbf{v}}_i)^\top \sigma(\mathbf{U}_i\mathbf{x}) \tag{41}$$

$$= \frac{1}{K}\sum_{i=1}^K \tilde{\mathbf{v}}_i^\top (\sigma(g(\mathbf{x})) - \sigma(\mathbf{U}_i\mathbf{x})) + \frac{1}{K}\sum_{i=1}^K \mathbf{v}_{anc}^{*\top}(\sigma(g(\mathbf{x})) - \sigma(\mathbf{U}_i\mathbf{x})). \tag{42}$$

Similar to Equation 17 and Equation 20, with probability $1 - \frac{\delta}{2}$, Equation 42 can be bound with

$$|z(\mathbf{x})| \leq \frac{1}{K} \sum_{i=1}^{K} |\tilde{\mathbf{v}}_i^\top (\sigma(g(\mathbf{x})) - \sigma(\mathbf{U}_i \mathbf{x}))| + \frac{1}{K} \sum_{i=1}^{K} |\mathbf{v}_{\text{anc}}^{*\top} (\sigma(g(\mathbf{x})) - \sigma(\mathbf{U}_i \mathbf{x}))| \tag{43}$$

$$\leq \frac{d\sqrt{\frac{1}{2} \log(4K/\delta)}}{K} \sum_{i=1}^{K} |(\sigma(g(\mathbf{x})) - \sigma(\mathbf{U}_i \mathbf{x}))| + \frac{d_{\text{anc}}\sqrt{\frac{1}{2} \log(4K/\delta)}}{K} \sum_{i=1}^{K} |(\sigma(g(\mathbf{x})) - \sigma(\mathbf{U}_i \mathbf{x}))|$$
$$\tag{44}$$

$$\leq \frac{d\sqrt{\frac{1}{2} \log(4K/\delta)}}{K} \sum_{i=1}^{K} |g(\mathbf{x}) - \mathbf{U}_i \mathbf{x}| + \frac{d_{\text{anc}}\sqrt{\frac{1}{2} \log(4K/\delta)}}{K} \sum_{i=1}^{K} |g(\mathbf{x}) - \mathbf{U}_i \mathbf{x}| \tag{45}$$

$$\leq \frac{(d + d_{\text{anc}})\sqrt{\frac{1}{2} \log(4K/\delta)}}{K} \sum_{i=1}^{K} |g(\mathbf{x}) - \mathbf{U}_i \mathbf{x}|. \tag{46}$$

Then similar to Equation 27, with probability $1 - \frac{\delta}{2}$, Equation 46 can be bound with

$$|z(\mathbf{x})| \leq \frac{(d + d_{\text{anc}})\sqrt{\frac{1}{2} \log(4K/\delta)}}{K^2} \sum_{i=1}^{K} \sum_{j \neq i} |(\mathbf{U}_j - \mathbf{U}_i)\mathbf{x}| \tag{47}$$

$$\leq \frac{(d + d_{\text{anc}})\sqrt{\frac{1}{2} \log(4K/\delta)}}{K^2} \sum_{i=1}^{K} \sum_{j \neq i} |(\mathbf{U}_j - \mathbf{U}_i)\mathbf{x}| \tag{48}$$

$$\leq \frac{(d + d_{\text{anc}})\sqrt{\frac{1}{2} \log(4K/\delta)}}{K^2} \sum_{i=1}^{K} \sum_{j \neq i} d\|\mathbf{x}\|_2 \sqrt{h \log(4hK^2/\delta)} \tag{49}$$

$$\leq \frac{\sqrt{2}}{2} d(d + d_{\text{anc}}) b \sqrt{h} \log(4hK^2/\delta). \tag{50}$$

Set the minimum of $\mathcal{L}_i$ closest to $\mathbf{w}_{\text{anc}}^*$ is $\mathbf{w}_{\text{anc},i}^*$. For $\sup_\alpha \mathcal{L}_i(\alpha \mathbf{w}_i + (1 - \alpha)\mathbf{w}_{\text{anc}}^*) < \epsilon$ holds with probability $1 - \delta$, then with probability $1 - \delta$ we have,

$$\sup_\alpha \mathcal{L}(\alpha \mathbf{w}_i + (1 - \alpha)\mathbf{w}_{\text{anc},i}^*) \leq \sup_\alpha \mathcal{L}(\alpha \mathbf{w}_i + (1 - \alpha)\mathbf{w}_{\text{anc}}^*) + \gamma \|\mathbf{w}_{\text{anc}}^* - \mathbf{w}_{\text{anc},i}^*\|_2^2 \tag{51}$$

$$\leq \epsilon + \gamma \Gamma^2. \tag{52}$$

Equation 51 is due to the assumption that $\mathcal{L}(\cdot)$ is $\gamma$-smooth. By Lemma 3.3, we have $d < \frac{d_{\epsilon + \gamma \Gamma^2}}{(1 - \delta)^{\frac{1}{S}}}$ with $S = hl + h$. Then $|z_\mathbf{x}(\alpha)|$ can be bounded as

$$|z_\mathbf{x}(\alpha)| \leq \frac{\sqrt{2h}b}{2(1 - \delta)^{\frac{2}{hl+h}}} d_{\epsilon + \gamma \Gamma^2}(d_{\epsilon + \gamma \Gamma^2} + d_{\text{anc}}) \log(4hK^2/\delta). \tag{53}$$

Now we turn to calculate the bound of the loss barrier $B_{loss}(\{\mathbf{w}_i\}_{i=1}^K)$. For the loss function $L(\cdot, y)$ is convex and 1-Lipschitz, similar to Equation 36, we have:

$$B_{\text{loss}}(\{\mathbf{w}_i\}_{i=1}^K) = \mathcal{L}\left(\frac{1}{K} \sum_{i=1}^{K} \mathbf{w}_i\right) - \frac{1}{K} \sum_{i=1}^{K} \mathcal{L}(\mathbf{w}_i) \tag{54}$$

$$= \mathbb{E}\left[L\left(f_{\frac{1}{K} \sum_{i=1}^K \mathbf{v}_i, \frac{1}{K} \sum_{i=1}^K \mathbf{U}_i}(\mathbf{x}), y\right) - \frac{1}{K} \sum_{i=1}^{K} L(f_{\mathbf{v}_i, \mathbf{U}_i}(\mathbf{x}), y)\right] \tag{55}$$

$$\leq \mathbb{E}\left[L\left(f_{\frac{1}{K} \sum_{i=1}^K \mathbf{v}_i, \frac{1}{K} \sum_{i=1}^K \mathbf{U}_i}(\mathbf{x}), y\right) - L\left(\frac{1}{K} \sum_{i=1}^{K} f_{\mathbf{v}_i, \mathbf{U}_i}(\mathbf{x}), y\right)\right] \tag{56}$$

$$\leq \mathbb{E}\left[\left|f_{\frac{1}{K} \sum_{i=1}^K \mathbf{v}_i, \frac{1}{K} \sum_{i=1}^K \mathbf{U}_i}(\mathbf{x}) - \frac{1}{K} \sum_{i=1}^{K} f_{\mathbf{v}_i, \mathbf{U}_i}(\mathbf{x})\right|\right], \tag{57}$$

Table 7: **Verification of transitivity of linear mode connectivity with less performed anchor models.** CIFAR-10. "Random init. Anchors" refers to that anchor models are randomly initialized models whose initializations are also different from the trained models. "Semi-trained Anchors" refers to that anchor models are trained for one epoch with less performed accuracy. It can be seen that when the anchor models are less performed ($\mathcal{A}(w_{anc})$s are low), the transitivity still holds that connectivity loss to the same anchor model can reduce connectivity barrier.

| Models | Metrics | Vanilla CE Loss | Connectivity Loss w/ Random Init. Anchors | Connectivity Loss w/ Semi-trained Anchors |
|--------|---------|-----------------|-------------------------------------------|-------------------------------------------|
| CNN | $\frac{\mathcal{A}(w_1)+\mathcal{A}(w_2)}{2}$ | 64.0±0.5 | 63.0±0.8 | 63.8±0.9 |
| CNN | $\mathcal{A}(w_{anc})$ | | 9.9±0.0 | 45.6±0.0 |
| CNN | $\mathcal{A}(\frac{w_{anc}+w_1}{2})$ | | 56.0±2.7 | 54.2±0.8 |
| CNN | $\mathcal{A}(\frac{w_1+w_2}{2})$ | 11.5±0.9 | 23.5±5.4 | 19.0±4.4 |
| CNN | Acc. Barrier | 0.821 | 0.626 (23.8%↓) | 0.702 (14.5%↓) |
| ResNet20 | $\frac{\mathcal{A}(w_1)+\mathcal{A}(w_2)}{2}$ | 66.7±0.9 | 67.4±1.3 | 69.0±0.2 |
| ResNet20 | $\mathcal{A}(w_{anc})$ | | 7.1±0.0 | 29.9±0.0 |
| ResNet20 | $\mathcal{A}(\frac{w_{anc}+w_1}{2})$ | | 38.3±4.1 | 42.4±1.2 |
| ResNet20 | $\mathcal{A}(\frac{w_1+w_2}{2})$ | 13.0±3.8 | 19.5±0.7 | 21.0±5.4 |
| ResNet20 | Acc. Barrier | 0.805 | 0.710 (11.8%↓) | 0.696 (13.5%↓) |

Table 8: **Comparison of computation cost to reach the target accuracies.** The computation cost is measured by the wall-clock time (minutes) during the implementation, and the less time, the less computation overhead. Settings: Tiny-ImageNet, non-IID hyper.=0.5, $M = 50$, $E = 3$. It can be seen that FedGuCci require less computation to reach the target accuracies.

| Methods | FedAvg | FedProx | FedDyn | FedRoD | MOON | FedLC | FedSAM | FedGuCci |
|---------|--------|---------|--------|--------|------|-------|--------|----------|
| Target Acc: 20% | 798m (×1.00) | 872m (×1.09) | 1091m (×1.37) | 759m (×0.95) | 848m (×1.06) | 652m (×0.82) | 748m (×0.94) | **578m (×0.72)** |
| Target Acc: 23% | / | 1173m (×1.00) | 1181m (×1.01) | 1337m (×1.14) | 2376m (×2.0267m (×1.08) | **752m (×0.64)** | | |
| Target Acc: 25% | / | 1413m (×1.00) | 1363m (×0.96) | / | 3649m (×2.58) | / | 1497m (×1.06) | **926m (×0.66)** |

Table 9: **Results under more heterogeneous settings.** Tinyimagnet, ResNet-18, $T = 50, M = 50, E = 3$.

| Methods | non-IID hyper.=0.1 | non-IID hyper.=0.05 |
|---------|--------------------|--------------------|
| FedAvg | 22.92±0.42 | 20.03±0.87 |
| FedDyn | 21.40±1.13 | 18.28±1.59 |
| FedSAM | 28.53±0.86 | 25.53±0.96 |
| FedGuCci | **30.33±0.35** | **26.39±0.41** |
| FedGuCci+ | **31.26±0.53** | **27.21±0.56** |

Table 10: **Experiments with smaller participation ratios.** Setting: K=100, CIFAR-10, non-IID $\alpha = 0.1$.

| Methods | Ratio = 5% | Ratio = 10% |
|---------|------------|-------------|
| Local | 21.80±1.49 | 26.82±0.09 |
| FedAvg | 62.54±0.28 | 64.15±0.11 |
| FedProx | 49.43±0.73 | 50.45±0.56 |
| FedRoD | 62.73±0.27 | 62.38±0.46 |
| FedLC | 62.47±0.57 | 63.57±0.13 |
| FedSAM | 61.92±0.44 | 63.99±0.41 |
| FedGuCci | **63.12±1.04** | **65.10±0.46** |
| FedGuCci+ | **63.61±0.24** | **64.57±0.44** |

where the expectation is with respect to the server dataset. Then use the bound of $z(\alpha)$, with probability $1 - \delta$, we have

$$B_{\text{loss}}(\{\mathbf{w}_i\}_{i=1}^K) \leq \frac{\sqrt{2h}b}{2(1-\delta)^{\frac{2}{hl+h}}} d_{\epsilon+\gamma\Gamma^2}(d_{\epsilon+\gamma\Gamma^2} + d_{\text{anc}}) \log(4hK^2/\delta). \tag{58}$$

□

## C  MORE RESULTS

In Table 7, we verify the transitivity of LMC under less performed anchor models, such as random initialization and semi-trained models. It can be seen that the transitivity stills holds regardless of the properties of anchor models. Though a better trained anchor model may lead to better transtivitiy.

In Table 8, we compare the computation costs of methods in terms of reaching a targeted accuracy.

In Table 9, we test our methods under more non-IID data, when in Table 10, we test our methods under smaller participation ratios. The results all show our methods are effective under these settings.

## D   MORE RELATED WORKS

**Linear Mode Connectivity.** Linear mode connectivity (LMC) refers to the phenomenon that there exists a loss (energy) barrier along the linear interpolation path of two networks, in the cases where i) the two networks have the same initialization and are trained on the same dataset but with different random seeds (data shuffles) or augmentations (Ainsworth et al., 2022); ii) the two networks are with different initializations but are trained on the same dataset (Entezari et al., 2022); iii) the two networks are the initial network and the final trained network (Vlaar & Frankle, 2022). In our paper, the transitivity of LMC can be applied to i), ii), and iii), and especially, the two trained models can have different initializations. Specifically, (Adilova et al., 2023) examines layer-wise LMC, and finds that there may be no barriers in the layer-wise manner. (Frankle et al., 2020) connects linear mode connectivity with the lottery ticket hypothesis and finds better connectivity can result in better pruning performances. (Vlaar & Frankle, 2022) studies the relationship between generalization and the initial-to-final linear mode connectivity. (Zhao et al., 2020) bridges mode connectivity and adversarial robustness. Some works try to extend mode connectivity beyond "linear", e.g., searching for a non-linear low-loss path (Draxler et al., 2018) or studying mode connectivity under spurious attributes (Lubana et al., 2023).

Studying the barriers in LMC is an important direction of LMC. Previous works find that there may be no barriers between different modes, but the connected regions may be non-linear (Draxler et al., 2018; Garipov et al., 2018). In (Garipov et al., 2018), the authors propose to find paths along modes by learning Polygonal chain and Bezier curve. Also, Nudged Elastic Band can also be used to find that connected paths (Draxler et al., 2018). In (Wortsman et al., 2021), the authors propose to learn connected but diverse low-loss subspaces for efficient ensembling. Our work about the transitivity of LMC is inspired by the previous works of learning connected paths. However, instead of learning diverse modes for ensembling, we aim to use the anchor model to improve the linear connectivity between two independent modes.

**Generalization of Federated Learning.** Generalization and personalization are two important goals of federated learning systems (Chen & Chao, 2022; Li et al., 2023a;b; Yuan et al., 2022). Previous works study and understand the property and nature of generalization in FL. In (Yuan et al., 2022), the authors rethink the previous definition of generalization by considering the data distributions of non-participated clients as the participation gap and propose a new data split method based on the insight. In the paper of FedRoD (Chen & Chao, 2022), the authors claim that generalization and personalization are not conflicted; instead, improving generalization is the basis for better personalization.

Some works aim to improve generalization from both the server and client sides. For the clients, sharpness-aware minimization methods are introduced at the local to find a flatter minimum of local solvers for better generalization (Caldarola et al., 2022; Qu et al., 2022). Global sharpness-aware minimization is also considered (Dai et al., 2023). In addition, previous literature seeks to tackle local heterogeneity to improve generalization, and methods like proximal terms (Li et al., 2020a), dynamic regularization (Acar et al., 2020), variance reduction (Karimireddy et al., 2020), logit calibration (Zhang et al., 2022), fixed classifier (Li et al., 2023b), and balanced loss (Chen & Chao, 2022) are devised. For the server, weighted aggregation approaches to de-bias local updates (Wang et al., 2020) or heterogeneity (Ye et al., 2023) can improve generalization. Recently, global weight shrinking that sets smaller aggregation weights has been studied for unleashing the potential of weight regularization in boosting the generalization of FL (Li et al., 2023a).

## E   LIMITATIONS AND BROADER IMPACTS

**Limitations.**   Though our methods are effective for improving the generalization of federated learning, they has limitations that it will introduce more computations than FedAvg. The introduced computations may cause more overhead of computing resources at the edge devices.

**Broader impacts.** The connectivity perspective of improving the generalization of federated will inspire more future works about model fusion. Model fusion has broad applications in large language models and other fields, and it can merge the abilities of multiple models and data resources. As far as we are concerned, our methods have no obvious negative impacts.

