# OpenReview forum: "Improving Group Connectivity for Generalization of Federated Deep Learning"
_ICLR.cc/2025/Conference — ICLR 2025 Conference Withdrawn Submission_

### Official Review · Reviewer_LCgW · 2024-10-26

**Soundness:** 4
**Presentation:** 3
**Contribution:** 3
**Rating:** 6
**Confidence:** 3

**Summary:**

This paper presents an interesting concept in FL often underexplored. Specifically, this paper leverages the concept of group connectivity, drawn from linear mode connectivity (LMC), to better fuse local models in parameter regions into a global generalized model, which could come with many benefits. The issue of model drift and heterogeneity seems to be tackled well. The authors validate the methods on vision and NLP datasets. Overall, the idea looks interesting.

**Strengths:**

1)	The paper provides a creative adaptation of LMC to FL, which is an underutilized idea in the FL domain.
2)	Well written paper and easy to follow.
3)	Sound theoretical foundations with supporting proofs to enhance the credibility.

**Weaknesses:**

I am particularly concerned with communication costs and computational overhead of the method when applied to large-scale FL scenarios. Below are few weaknesses:

1) Regarding the communication costs, I agree that there are no additional costs compared to FedAvg, but this remains an issue. There have been many advancements in communication efficiency over FedAvg in recent years. To mention a few recent works, DisPFL[1] and SSFL[2] have demonstrated better performance than FedAvg, even at very high sparsity levels. I believe the baselines are insufficient and suggest comparing the method with at least these sparse baselines on a few vision datasets to evaluate whether the performance gains justify the communication cost to validate its worth.

2) The authors did not consider incorporating techniques to prune or compress unimportant weights, which could reduce communication overhead without sacrificing performance. While the core idea of the paper is appreciated, there is room for optimization and further improvements in terms of communication efficiency.

2) Although the idea is novel, it’s a lot of computations and could pose practical challenge in large-scale FL applications. Authors should think of addressing this in the future works so that this method can be practically applied.

References:

[1]  https://doi.org/10.48550/arXiv.2206.00187

[2] https://doi.org/10.48550/arXiv.2405.09037

**Questions:**

1.	Can authors provide insights into storage challenges in very large-scale scenarios?  Furthermore, I would be happy if I could see 1-2 experiments on large clients (M > 500) or (M > 1000) if possible.
2.	Theoretically, can authors intuitively explain how the model behaves and challenges when applied to models like GPT-3? Is there a way to expand this for such large models in the near future?
3.	Could the authors elaborate more on how the method would behave in low-bandwidth, high-latency federated learning environments? Would there be trade-offs in performance?
4. Please respond to the weaknesses above.

---

### Official Review · Reviewer_6k6G · 2024-11-04

**Soundness:** 2
**Presentation:** 2
**Contribution:** 2
**Rating:** 3
**Confidence:** 3

**Summary:**

FedGuCci(+) enhances federated learning generalization by improving group connectivity, inspired by linear mode connectivity, to better fuse local models and achieve stronger performance across diverse tasks and architectures.

**Strengths:**

This paper adapts a connectivity perspective to use LMC improving the connectivity among the local models. It shall be a novel tool in the analysis of FL.

**Weaknesses:**

W1. The comparison with prior work is limited

W2. The proposed algorithms, including FedGuCci, do not have theoretical convergence guarantees.

W3. The comparisons and conclusions may not have sufficient evidence.

W4. The evaluation needs to be enhanced.

**Questions:**

Q1: In Section 2.3, the way your method leverages the global model is different from FedProx, but it’s unclear what specific advantages your approach offers. While your method integrates the connectivity loss and replays N historical global models, FedProx only uses the current global model for regularization. A more detailed explanation of why your method is superior would be helpful.

Q2: The proposed algorithms do not have theoretical convergence guarantees. Many of the methods you compare against do provide convergence analyses, and including such an analysis would be essential for the theoretical foundation of your work.

Q3:  When discussing group connectivity varying K, the range of client numbers is too small to reflect the truth in practice. The left plot in Figure 4 may not support the author’s conclusion that “the increase of barriers may converge to a point lower than vanilla training”. If it finds the transitivity of group connectivity may be weakened for larger K, an interpretation of the reason should be provided.

Q4. If FedGuCci+ is designed to enhance FedGuCci by incorporating additional techniques, it’s important to evaluate the individual contribution of each technique to the overall improvement, like conducting ablation studies.

---

### Official Review · Reviewer_3d9k · 2024-11-04

**Soundness:** 2
**Presentation:** 3
**Contribution:** 2
**Rating:** 3
**Confidence:** 4

**Summary:**

The paper proposed FedGuCci, an algorithm to improve merging of local client models in federated learning by ensuring that the client models are in the same loss basin, motivated by findings in the linear mode connectivity (LMC) literature. To do so, authors propose to add a connectivity loss to the standard client objective which tries to ensure that the client's local model is mode connected to an anchor model. The authors theoretically motivate their approach by proving the transitivity property of LMC when merging two layer neural networks. This is followed by experiments on practical FL training tasks, which shows that the proposed FedGuCci and it's extension FedGuCci+ can outperform FedAvg and other baselines.

**Strengths:**

1. The paper is well-structured and easy to read and intuitively the idea of using anchor models to improve LMC cross the client local models makes sense.

2. Experimental results look promising and show that the proposed FedGuCci and FedGuCci+ can outperform vanilla FedAvg and other baselines across a wide range of settings.

3. Ablation studies are provided showing that the proposed algorithms are generally robust and require lower computation cost to reach a target accuracy compared to FedAvg and other baselines.

**Weaknesses:**

1.  **Lemma 3.3 statement and implication**

* Firstly, I don't understand what is random about $D_\\epsilon(w_{anc}^*)$ defined in Lemma 3.3. If  $w_{anc}^*$ is a deterministic anchor model (as defined later in Theorem 3.5 and Theorem 3.8) then $D_\\epsilon(w_{anc}^*)$ is also deterministic so the bound on the probability of $D_\\epsilon(w_{anc}^*)$ does not make sense to me. Intuitively, I believe Lemma 3.3 is trying to bound the probability of LMC between a deterministic anchor model $w_{anc}^*$ and a random model $w$ such that $||w-w_{anc}^*|| \leq d/2$. So the probability here is over some uniform distribution of $w$ and not $w_{anc}^*$. If so, Lemma 3.3 should be re-written to clarify this.

* There is no understanding of how large $d_{\epsilon}$ can be. For instance if $d_{\epsilon} \geq d$ then the probability bound is just vacuous. Therefore authors need to either assume with some justification or prove that $d_\epsilon < d$ for the bound to make sense.

2. **Theorem 3.5 and 3.8 statement and implications.**

* It is not clear to me why we need randomness in these theorem statements. Suppose we are given three models $w_{anc}, w_1, w_2$ such that i) $w_{anc}$ and $w_1$ are LMC ii) $w_{anc}$ and $w_2$ are LMC  and iii)$||w_{anc} - w_1|| \leq d$ and $||w_{anc} - w_2|| \leq d$. Why can't we use just these assumptions to bound the loss barrier between $w_1$ and $w_2$? In other words why do we need the additional assumption that $w_1$ and $w_2$ are sampled from the uniform distribution?

* Following up on my point in Weakness 1), authors need to show that $d_\epsilon$ is bounded for the bound in Eq. (6) and Eq. (10) to not be vacuous. In addition, authors also need to show that $\delta > 0$ that this there is a non-zero probability that $w_1$ and $w_2$ are sampled from the uniform distribution and are also LMC with $w_{anc}$.

* Currently I don't see a dependence on $d$ in the bounds in Eq. (6) and Eq. (10) which is a bit surprising to me since $d$ is defined in the Theorem statement.

3. **Inconsistent experimental results and performance of baselines**

* It appears to me that the authors have either not implemented FedProx correctly or have not tuned the regularization parameter $\mu$ in FedProx correctly. If we set $\mu \rightarrow 0$ then FedProx just becomes same as FedAvg so the performance of FedProx should at least be as good as FedAvg. Therefore it is surprising to see the consistently poor performance of FedProx across all the experiments.

* In Table 2, for the Tiny ImageNet column with Non-IID hyper. being 100, the performance of SCAFFOLD seems inconsistent with previous results.

* In Table 3, for the STS-B column, the performance of FedDyn seems inconsistent with previous results.

* Authors should provided information on how they tuned hyperparameters for the experiments and also provide graphs showing test accuracy vs number of rounds for all the baselines. Currently only the final accuracy numbers is reported for each of the experiments

*  Why is FedDyn and SCAFFOLD not compared against in Table 4 and Table 5?

**Questions:**

1. The definition of the connectivity loss in Eq. (7) requires an integration over $\alpha$ which I believe is not possible to implement practically. The authors should comment on how they approximate this loss in practice.

2. What is the neural network used in Figure 2?

3. Can we extend the definition of group connectivity to work with general weights instead of fixing the weights as $1/K$ for each of the group models?

4. For overparameterized models (every global optimum is also a local optimum), it appears that the value of $\Gamma$ in Definition 3.7 would be zero, implying no effect of heterogeneity on the bound in Theorem 3.8. Can the authors comment on this?

**Typos**
1. There appears to be a typo in Line 95 in the statement "refer to each client's distribution $D_i$".
2. It should be $D^t$ and not $D$ in Line 113.

---

### Note · Authors · 2024-11-13

I have read and agree with the venue's withdrawal policy on behalf of myself and my co-authors.